# MRI-based microthrombi detection in stroke with polydopamine iron oxide

Charlène Jacqmarcq[1], Audrey Picot[1], Jules Flon[1], Florent Lebrun [1],
Sara Martinez de Lizarrondo[1], Mikaël Naveau[2], Benoît Bernay[3], Didier Goux[4],
Marina Rubio [1], Aurélie Malzert-Fréon [5], Anita Michel [6], Fabienne Proamer[6],
Pierre Mangin [6], Maxime Gauberti [1,7], Denis Vivien [1,8] ✉ &
Thomas Bonnard [1] ✉

In acute ischemic stroke, even when successful recanalization is obtained, downstream microcirculation may still be obstructed by microvascular thrombosis, which is associated with compromised brain reperfusion and cognitive decline. Identifying these microthrombi through non-invasive methods remains challenging. We developed the PHySIOMIC (Polydopamine Hybridized Self-assembled Iron Oxide Mussel Inspired Clusters), a MRI-based contrast agent that unmasks these microthrombi. In a mouse model of thromboembolic ischemic stroke, our findings demonstrate that the PHySIOMIC generate a distinct hypointense signal on $T_2$*-weighted MRI in the presence of microthrombi, that correlates with the lesion areas observed 24 hours post-stroke. Our microfluidic studies reveal the role of fibrinogen in the protein corona for the thrombosis targeting properties. Finally, we observe the biodegradation and biocompatibility of these particles. This work demonstrates that the PHySIOMIC particles offer an innovative and valuable tool for non-invasive in vivo diagnosis and monitoring of microthrombi, using MRI during ischemic stroke.

Acute ischemic stroke (AIS) results from the occlusion of a cerebral artery due to thrombus formation and is the leading cause of acquired disability among adults and the second cause of dementia worldwide. Current strategies for recanalization of the occluded artery encompass thrombolysis and mechanical thrombectomy (MT). Thrombolysis consists of the administration of recombinant tissue-type Plasminogen Activator (rtPA: Alteplase) or Tenecteplase (TNK) within a therapeutic time window of 4.5 hours post-onset of symptoms, which may be extended following specific imaging criteria[1]. The utilization of rtPA or

TNK remains limited, constituting <20% of hospital-admitted AIS patients[2], owing to its manifold contraindications and noteworthy side effects, particularly hemorrhagic transformations. Following a series of clinical trials from 2015 onwards that substantiate the efficacy of recanalization through MT[3–6], this approach has now become integral to the standard care for AIS patients. MT can be employed either in tandem with or as a potential alternative to rtPA-mediated thrombolysis, broadening the scope of AIS patient eligibility for treatment. However, microembolic signals are found in patients treated with

[1]Normandie University, UNICAEN, Université Caen Normandie, INSERM UMR-S U1237, Physiopathology and Imaging of Neurological Disorders (PhIND), GIP Cyceron, Institute Blood and Brain @ Caen-Normandie (BB@C), Caen, France. [2]Normandie University, UNICAEN, Université Caen Normandie, CNRS UMS 3408 Caen, France. [3]Normandie University, UNICAEN, Université Caen Normandie, SF 4206 ICORE, Plateforme Proteogen, Caen, France. [4]Normandie University, UNICAEN, Université Caen Normandie, US EMerode, CMAbio3: Centre de Microscopie Appliquée à la Biologie, Caen, France. [5]Normandie University, UNICAEN, Université Caen Normandie, EA 4258, CERMN: Centre d'études et de recherche sur le médicament de Normandie, Caen, France. [6]University of Strasbourg, INSERM, EFS Grand-Est, BPPS UMR-S1255, FMTS, F-67065 Strasbourg, France. [7]Centre Hospitalier Universitaire Caen, Department of Diagnostic Imaging and Interventional Radiology, Caen, France. [8]Centre Hospitalier Universitaire Caen, Department of Clinical Research, Caen, France.
✉e-mail: vivien@cyceron.fr; bonnard@cyceron.fr

thrombolytics[7,8], and microfragmentations of the proximal thrombus—denoted as microthrombi—manifesting as distal microvascular migration are observed in 14 to 20% of patients undergoing MT[9]. The persistence of microthrombi despite a successful recanalization of the occluded artery is associated with an increased risk of dementia and unfavorable patient outcomes[10–12]. In addition, animal models have reinforced the idea that reperfusion within the microvasculature can positively impact stroke recovery[13] and in 2022, the CHOICE Clinical Trial investigated the potential benefits of implementing thrombolysis subsequent to successful thrombectomy[14].

Considering these diverse factors, it is imperative to ensure accurate visualization of microthrombi during AIS and to follow the recanalization of occluded arteries. Regrettably, the visualization of microthrombi within cerebral vascularization using conventional imaging techniques remains challenging due to their diminutive size. This challenge persists despite recent advances to visualize distal emboli, such as high-resolution diffusion-weighted imaging (DWI) in magnetic resonance imaging (MRI) and transcranial Doppler methods[15]. Although contrast agents have been recently synthesized to enhance thrombus visualization in both MRI and computed tomography (CT)[16,17], their sensitivity remains inadequate to detect microthrombi effectively.

Superparamagnetic iron oxide nanoparticles (SPIO) represent a distinctive class of contrast agents used in MRI applications. These nanoparticles have gained approval for human administration, primarily for patients dealing with iron-deficient anemia (Feraheme®)[18]. They are also being developed for lymph node imaging[19], in hyperthermia for cancer therapy[20], and are under investigation in numerous clinical trials. With their extended half-life, minimal toxicity, and biocompatible attributes, these nanoparticles have captured significant attention as thrombus-targeted imaging candidates. However, the practical application of SPIO is constrained by particle size, limiting their potential for effective target-specific binding and robust MRI signal[21].

Polydopamine, a bio-inspired polymer, emerges as a noteworthy contender within the domain of bioengineering and biomedicine. Resulting from dopamine polymerization under alkaline conditions and in the presence of oxygen, the polydopamine material has been extensively used over the past decade for the synthesis of anti-cancer drug delivery nanomedicines and diagnostic tools[22–24]. Beyond its biocompatibility, polydopamine offers a range of attractive features suitable for contrast agent design. These include (i) a pronounced affinity for metal oxide surfaces, (ii) a facilitated surface functionalization to targeting moieties via Michael addition, and (iii) straightforward self-assembly synthesis protocols[25].

In this study, we employ polydopamine for the encapsulation of commercially available SPIO and the subsequent detection of microthrombosis using MRI in experimental models of AIS in mice.

## Results

### PHySIOMIC synthesis and characterization

We formulated clusters composed of SPIO aggregates within a polydopamine matrix through a self-polymerization process, outlined in Fig. 1a. The assembly of SPIO clusters within the polydopamine matrix occurred during the polymerization step under alkaline conditions. By subjecting the solution to sonication, we generated microparticles termed Polydopamine Hybridized Self-assembled Iron Oxide Mussel Inspired Clusters (PHySIOMIC). The clusterization of SPIO into PHySIOMIC was confirmed through the measurement of the hydrodynamic diameter (HD) obtained via dynamic light scattering (DLS) (Fig. 1b). DLS analysis of PHySIOMIC revealed an average HD of $753.7 \pm 47.5$ nm with a polydispersity index (PDI) of $0.22 \pm 0.19$, while SPIO exhibited an average HD of $78.5 \pm 11.3$ nm with a PDI of $0.21 \pm 0.04$ (Table 1). Notably, the size of PHySIOMIC was observed to be tenfold that of SPIO, yet iron quantification in equivalent volumes and

dilutions of SPIO or PHySIOMIC demonstrated that the majority of the iron content is conserved during the synthesis process of clusters ($0.481 \pm 0.024$ mg mL$^{-1}$ for SPIO and $0.417 \pm 0.017$ mg mL$^{-1}$ for PHySIOMIC). Furthermore, the DLS assessment revealed a negative zeta potential for PHySIOMIC ($-36.4 \pm 2.4$ mV), consistent with prior experimental observations in the literature of polydopamine particles[24]. In contrast, SPIO exhibited a less negative zeta potential ($-11.1 \pm 1.6$ mV, $p$ value = 0.0001), which can be attributed to the presence of carboxyl groups on the dextran periphery of the SPIO particles[26].

The PHySIOMIC structure was visualized using Transmission Electron Microscopy (TEM), confirming the clustering of SPIO within the polydopamine matrix (Fig. 1c). Optical microscopy was employed to observe particle distribution and ensure the absence of aggregation in a mannitol buffer (Fig. 1d). The relaxation rates and relaxivity (r2*) of PHySIOMIC and SPIO were measured in phantoms using 7 T magnetic resonance imaging (MRI). PHySIOMIC exhibited higher relaxivity compared to SPIO particles (r2* = 388.3 mM$^{-1}$ s$^{-1}$ versus 153.7 mM$^{-1}$ s$^{-1}$, $n = 1$), while at similar iron content (Fig. 1e). This observation translated into approximately twofold higher relaxivity value than that of SPIO particles for $T_2$*-weighted imaging (Fig. 1f, Table 1). This outcome underscores the heightened sensitivity of PHySIOMIC to $T_2$*-weighted MRI imaging, even at low concentrations, attributed to the increased magnetic moment conferred by larger particles[27]. Moreover, r1 and r2 relaxivity values were computed for both types of particles, revealing a similar r2 value but a reduced r1 value for PHySIOMIC (Supplementary Fig. 1, Table 1), thus indicating a notably higher r2/r1 ratio.

To ensure the safety of particle injection, we first conducted studies on in vitro degradation, non-hemolysis, and non-impact clot lysis. The degradation assessment involved the exposure of both PHySIOMIC and SPIO particles to diverse buffers for a duration of 7 days, at 37 °C. No degradation was observed for both PHySIOMIC and SPIO particles in the PBS buffer. However, a distinct degradation of iron oxides became apparent in the citrate buffers for SPIO particles, resulting in a transition from a brown color to a clear liquid (Supplementary Fig. 2a). Hydrogen peroxide generates free radicals capable of degrading polydopamine. Consequently, for PHySIOMIC particles, the most pronounced degradation was observed in the citrate with hydrogen peroxide buffer, as well as in artificial lysosomal fluid (ALF), where the near-infrared absorbance value approached zero, as for SPIO (Supplementary Fig. 2b). As a result, the UV-Vis results underscore the degradation of polydopamine within the PHySIOMIC particles across various buffers. To ascertain non-hemolysis upon injection into the bloodstream, the particles were incubated at increasing iron concentrations, reaching up to 80 μL[Fe] mL$^{-1}$, alongside erythrocytes isolated from the blood of healthy volunteers (Supplementary Fig. 3a). The calculated hemolysis rates remained very low, with no condition exceeding a 5% hemolysis threshold (Supplementary Fig. 3b), a critical hemolytic ratio threshold for biomaterials according to ISO/TR 7406. One of the major risks associated with particle injection following an ischemic stroke is the potential interference with clot lysis, either natural or induced by thrombolytic agents. To address this, a clot lysis test was performed. The particles were introduced into a well containing human plasma, where a clot was initiated through the introduction of calcium and subsequently lysed with the addition of rtPA (Supplementary Fig. 4a). Notably, no significant distinction was measured between wells containing PHySIOMIC particles and those comprising rtPA alone ($p$ value = 0.0769 for tPA 1 nM doses and $p$ value = 0.8153 for tPA 5 nM doses; Supplementary Fig. 4b).

### MRI of microthrombi in an ischemic stroke mouse model

We then employed a magnetic resonance molecular imaging strategy to detect the presence of microthrombi in a murine model of thromboembolic ischemic stroke, using SPIO and PHySIOMIC particles, which exhibited favorable relaxivity as confirmed through

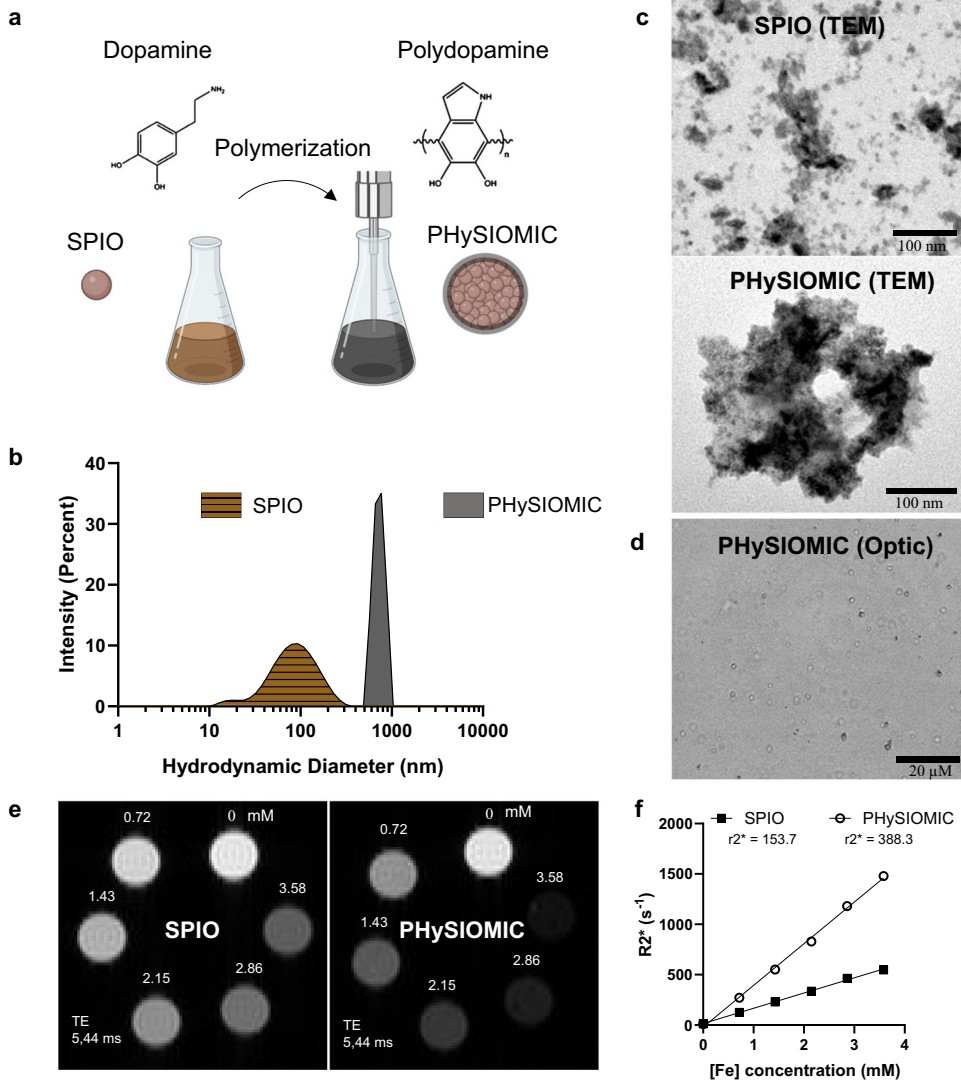

**Fig. 1 | PHySIOMIC synthesis and physico-chemical characteristics. a** Schematic diagram illustrating the PHySIOMIC synthesis process, which involves a simple polymerization step of dopamine to facilitate the clusterization of SPIO into PHySIOMIC. Created with BioRender.com. **b** DLS analysis of PHySIOMIC and SPIO, providing the mean hydrodynamic diameter ($n = 3$ particles preparations). **c**, **d** Morphological and core size observations conducted using transmission electron microscopy (TEM) showcasing the clusterization of SPIO into PHySIOMIC, and optical microscopy demonstrates the proper dispersion PHySIOMIC in a mannitol solution ($n = 3$ particles preparations). **e**, **f** MRI scans and relaxivity measurements obtained from a T2*-weighted sequence of PHySIOMIC and SPIO. Supplementary values of the DLS analysis are given in Table 1, and Supplementary Fig. 1. offers supplementary information on the r1 and r2 relaxivities ($n = 1$ particle preparation). Source data are provided as a Source Data file.

characterization studies. In this ischemic stroke model, thrombin was injected into the middle cerebral artery (MCA) to induce thrombus formation at the bifurcation of the artery. Importantly, the thrombin also travels through the downstream microvasculature, prompting the development of microthrombi alongside the proximal thrombus[28]. Following the induction of the middle cerebral artery occlusion (MCAO) model, $T_2$*-weighted MRI scans were performed before and after intravenous injection of SPIO or PHySIOMIC (both solutions at 1 mg [Fe] kg$^{-1}$), and lesion sizes were assessed 24 hours after occlusion (Fig. 2a). Intravenous injection of SPIO post-stroke did not result in any discernible hypointensity within the cerebral cortex. Conversely, the administration of PHySIOMIC yielded distinct hypointensity across the ischemic cerebral cortex, notably prominent in the penetrating arterioles, highlighting the microthrombi initiated by thrombin injection into the distal circulation (Fig. 2b). These observations are further supported by signal quantification (Fig. 2c), including the measurement of the hypointensity voxel volume (signal void) within the cortex significantly different before and after PHySIOMIC injection (Signal

Void of $0.40 \pm 0.32$ mm$^3$ before versus $2.57 \pm 1.20$ mm$^3$ after PHySIOMIC injection, $n = 8$, $p$ value = 0.0078) and no differences are observed with SPIO (Signal Void of $0.15 \pm 0.32$l mm$^3$ compared to $0.20 \pm 0.11$ mm$^3$ after SPIO injection, $n = 5$, $p$ value = 0.555).

It is noteworthy that the hypointensity induced by PHySIOMIC becomes apparent on MRI scans just a few minutes after their intravenous injection (Supplementary Movie 1) and that the location of hypointensities, evident in $T_2$*-weighted images, corresponds to the cortex impacted by the ischemia (Fig. 2d). A significant correlation was measured between the lesion sizes observed in $T_2$-weighted images 24 hours after occlusion and the cerebral volumes featuring PHySIOMIC signals (Fig. 2e), quantified as the signal void area (Pearson r(6) = 0.9329, $p$ value = 0.0007). Overlaying both signals in the 3D reconstructed brain (Fig. 2f) elucidates that the regions comprising microthrombi (30 minutes post-occlusion) align with those of ischemic lesions (24 hours post-occlusion). We can conclude that, in this model, PHySIOMIC particles constitute a diagnosis tool to reveal microthrombi and to predict the size of the ischemic

**Table 1 | Comparison of physico-chemical properties of SPIO and PHySIOMIC**

| | SPIO | PHySIOMIC | p value |
|---|---|---|---|
| Hydrodynamic diameter (nm) ± SD | 78.5 ± 11.3 | 753.7 ± 47.5 | $p < 0.0001$ |
| Polydispersity index ± SD | 0.22 ± 0.04 | 0.22 ± 0.19 | $p = 0,9831$ |
| Zeta potential (mV) ± SD | −11.1 ± 1.6 | −36.4 ± 2.4 | $p = 0.0001$ |
| Iron concentration (mg mL$^{-1}$) ± SD | 0.481 ± 0.024 | 0.417 ± 0.017 | $p = 0.0189$ |
| Relaxivity r1 (mM$^{-1}$ s$^{-1}$) | 0.8 | 0.5 | |
| Relaxivity r2 (mM$^{-1}$ s$^{-1}$) | 121.1 | 119.6 | |
| Relaxivity r2* (mM$^{-1}$ s$^{-1}$) | 153.7 | 388.3 | |

Hydrodynamic diameter (HD), polydispersity index (PDI), and zeta potential were determined using dynamic light scattering (DLS) analysis. Iron concentration was measured through a ferrozine assay. Relaxivity values (r1, r2, r2*) were calculated from MRI signal. Except for relaxivity values ($n = 1$ particles preparation), each experiment was performed in triplicate and expressed as mean ± SD ($n = 3$ particles preparation, Student $t$ test, two-sided). Source data are provided as a Source Data file.

lesions. Microthrombi could be identified via platelet marker CD41 immunolabelling in the ischemic area on histological analysis of brains harvested just after MRI acquisition, 1 hour after the stroke induction. The PHySIOMIC particles could be detected via light reflection from polydopamine positioned around the microthrombi on the luminal side of blood vessels (Supplementary Fig. 5a, b). Perls staining confirmed this specific localization of the particles, stained blue due to their iron oxide composition (Supplementary Fig. 5c). To further investigate the localization of the PHySIOMIC within the microthrombi, we performed transmission electronic microscopy (TEM) observation of the brain section, and we observed the iron oxide clusters close to degranulated platelets (Fig. 2g and Supplementary Fig. 6). In all observations of the histological sections, the PHySIOMIC particles were identified inside or at the surface of microthrombi, without noticeable accumulation in other parts of the brain.

Subsequently, the MRI signal emitted by PHySIOMIC was assessed over a 24-hour period. Following a single intravenous injection after occlusion, MRI scans were conducted every 6 hours for 24 hours to determine the duration for which the signal remained detectable post-injection of the contrast agent. The MRI scans depict a noticeable hypointensity at 10 minutes and 6 hours after occlusion, which gradually diminishes and then sharply fades at the 24-hour mark (Supplementary Fig. 7a, b). Quantitative analysis of the signal void reveals a significant difference in the volume of hypointense voxels at 10 minutes compared to baseline (Signal Void of 2.85 ± 1.03 mm$^3$ at 10 minutes compared to 0.21 ± 0.17 mm$^3$ at baseline, $n = 5$, $p$ value = 0.0066) and 6 hours (Signal Void of 2.02 ± 1.95 mm$^3$ at 6 hours compared to baseline, $p$ value = 0.0203). The decreasing pattern reaches 0.55 ± 0.73 mm$^3$, 24 hours after injection, similar ($p$ value > 0.999) to the baseline values (Supplementary Fig. 7c).

To investigate the presence of a sex-related effect, we concurrently administered PHySIOMIC particles to label microthrombi in a group of male mice and a group of female mice (Supplementary Fig. 7d). Quantification of the hypointensity observed in the MRI images showed no significant differences in microthrombi between male and female mice (Supplementary Fig. 7e). Subsequent phases of the study focused exclusively on male mouse cohorts.

We further compared the results obtained after PHySIOMIC injection in this first ischemic stroke model induced via thrombin injection in the MCA (referred to as 'thrombin model'), to a second ischemic stroke model in which thrombosis was induced in situ via local deposition on the MCA of a filter paper soaked with aluminum

chloride (Fig. 3). In this second model (referred to as the 'AlCl$_3$ model'), a unique clot is formed precisely at the area in contact with AlCl$_3$ and no embolization in the microcirculation occurs. This experiment had two main objectives. Firstly, it sought to demonstrate that there are no MRI modalities capable of distinguishing between the two models, with a significant pathological difference, without the use of contrast agents. Secondly, it aimed to establish the specificity of the particles for microthrombi, as opposed to a result of other events that occur during strokes. Consequently, we obtained $T_1$-weighted, $T_2$, $T_2$*-weighted, and diffusion-weighted (DWI) MRI scans without contrast injection, and we observed no visible differences between the two models in these sequences (Fig. 3a). The only differentiation became apparent in the $T_2$*-weighted sequence following the injection of PHySIOMIC particles, enabling the observation of microthrombi in the thrombin model. Notably, perfusion-weighted imaging with Gadolinium injection and cerebral angiography also failed to reveal any disparities ($n = 5$, $p$ value = 0.6674 and $p$ value > 0.999, respectively) between the two models (Fig. 3b). Except post-PHySIOMIC injection MRI, the visualization of cerebral tissues through immunohistochemistry was the sole method enabling the identification of microthrombi in the thrombin model and their absence in the AlCl$_3$ model (Fig. 3c). The images observed after immunochemistry reveal blue-marked platelets (CD41) and yellow-marked fibrin in the ischemic cortex following the thrombin model, indicative of the presence of microthrombi. These structures are absent in the AlCL$_3$ model.

**Monitoring microthrombi thrombolysis using PHySIOMIC**

The thrombin-induced stroke model is widely employed to assess the effectiveness of novel thrombolytic agents[29,30]. Our aim was to evaluate the efficacy of the early administration (30 minutes after stroke onset) of recombinant tissue-type Plasminogen Activator or rtPA (Alteplase®)—the gold standard for thrombolytic therapy in ischemic stroke patients—on microthrombi, using monitoring of microthrombi with PHySIOMIC on $T_2$*-weighted MRI scans (Fig. 4a).

While images acquired before rtPA treatment exhibited a strong hyposignal on $T_2$*-weighted images, a significantly lower signal was observed after rtPA treatment (Signal Void of 6.50 ± 3.42 mm$^3$ before tPA treatment versus 2.72 ± 3.32 mm$^3$ after treatment, $n = 7$, $p$ value < 0.0001), in accordance to a reduction of the microthrombosis (Fig. 4b, c). Mice treated with saline displayed no measurable disparities in hyposignals (Signal Void of 7.95 ± 7.11 mm$^3$ pre-treatment and 8.042 ± 7.67 mm$^3$ post-treatment, $n = 7$, $p$ value = 0.9732) (Fig. 4d). Accordingly, magnetic resonance angiographies demonstrate a restoration of the blood flow in the MCA and downstream vasculature in rtPA-treated mice. Previous studies using the thrombin model have demonstrated that, similar to ischemic stroke in humans, spontaneous recanalization of the MCA may occur within 24 hours after stroke onset in the absence of any treatment. However, the recanalization is faster and more frequent when rtPA is administered early after stroke onset[31]. Our results are in accordance with these findings, as the follow-up angiography study shows that the early administration of rtPA significantly increases the MCA recanalization at 24 hours ($n = 6$ saline-treated and $n = 7$ tPA-treated, $p$ value = 0.0003) (Fig. 4e, f), supporting a correlation between blood flow recovery and the reduction of microthrombosis. In line with this experiment, we also aimed to determine the extent of microthrombi at various time points following occlusion initiation, and administered PHySIOMIC at varying intervals post-occlusion, with MRI scans acquired immediately (Supplementary Fig. 8a, b). While the signal void was consistent with previous observations when measured at early time points after stroke onset (30 minutes and 2 hours post-occlusion), it almost completely disappears at 6 hours post-occlusion (Signal Void of 1.43 ± 1.46 mm$^3$ 30 minutes post-occlusion versus 0.14 ± 0.09 mm$^3$ at 6 hours, $n = 6$, $p$ value = 0.0252) (Supplementary

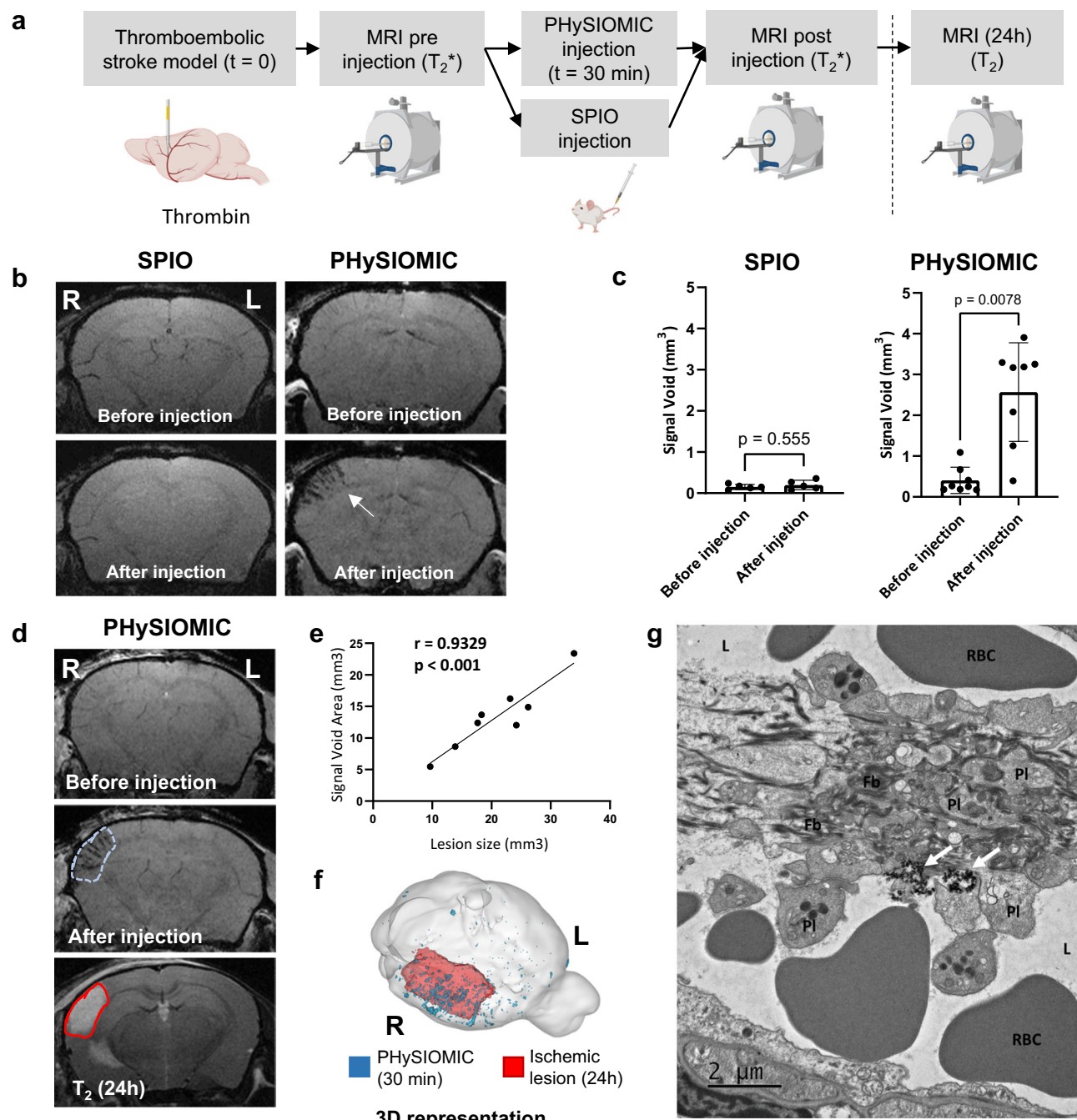

**Fig. 2 | PHySIOMIC reveals microthrombosis during the acute phase of stroke in a thromboembolic stroke model, evidenced by a hyposignal that correlates with lesion size at 24 h. a** Protocol illustration of intravenous injection of PHy-SIOMIC or SPIO in a thrombin-induced stroke model, followed by acquisition of T2*-weighted images before, 30 minutes, and 24 hours after PHySIOMIC injection. Created with BioRender.com. **b** T2*-weighted MRI scans demonstrating the hypointense signal (arrow) induced by PHySIOMIC particles in the brain cortex, while no hypointense signal is observed with SPIO. **c** Quantification of signal void in mm3 before and after PHySIOMIC ($n = 8$ animals, paired $t$ test, two-sided) or SPIO injection ($n = 5$ animals, paired $t$ test, two-sided). Data are presented as mean ± SD. **d** The signal void area observed 30 min after the occlusion in a T2*-weighted

sequence (blue dotted line) is predictive of the lesion size observed at 24 h after occlusion in a T2-weighted sequence (red continuous line). **e** Signal void area measures show a significant correlation with lesion size at 24 h ($n = 8$ animals, Pearson correlation coefficient, two-sided). Source data are provided as a Source Data file. **f** 3D representation of PHySIOMIC-induced hypointense signal and lesion hypersignal in MRI. R and L indicate right and left side of the presented brain. **g** TEM was performed on brain tissue collected after PHySIOMIC injection to visualize the presence of PHySIOMIC in the brain (observations in one animal). The images show PHySIOMIC particles (arrows) positioned around the clot composed of platelets (Pl) and fibrin (Fb) in the vessel lumen (L). RBC: Red Blood Cell. More images are shown in Supplementary Fig. 7.

Fig. 8c). These events coincide with cerebral reperfusion, as indicated by the increase in the angiographic score which supports that this progressive signal disappearance is caused by the clearance of the microthrombi rather than in situ metabolization of the PHy-SIOMIC (Supplementary Fig. 8d, e).

## Passive targeting of microthrombi via protein corona

We investigated how the PHySIOMIC was able to target thrombi and microthrombi without the need for specific ligands. This prompted us to direct our focus towards the protein corona (PC) which is formed around particles upon contact with blood. The PC plays a pivotal role

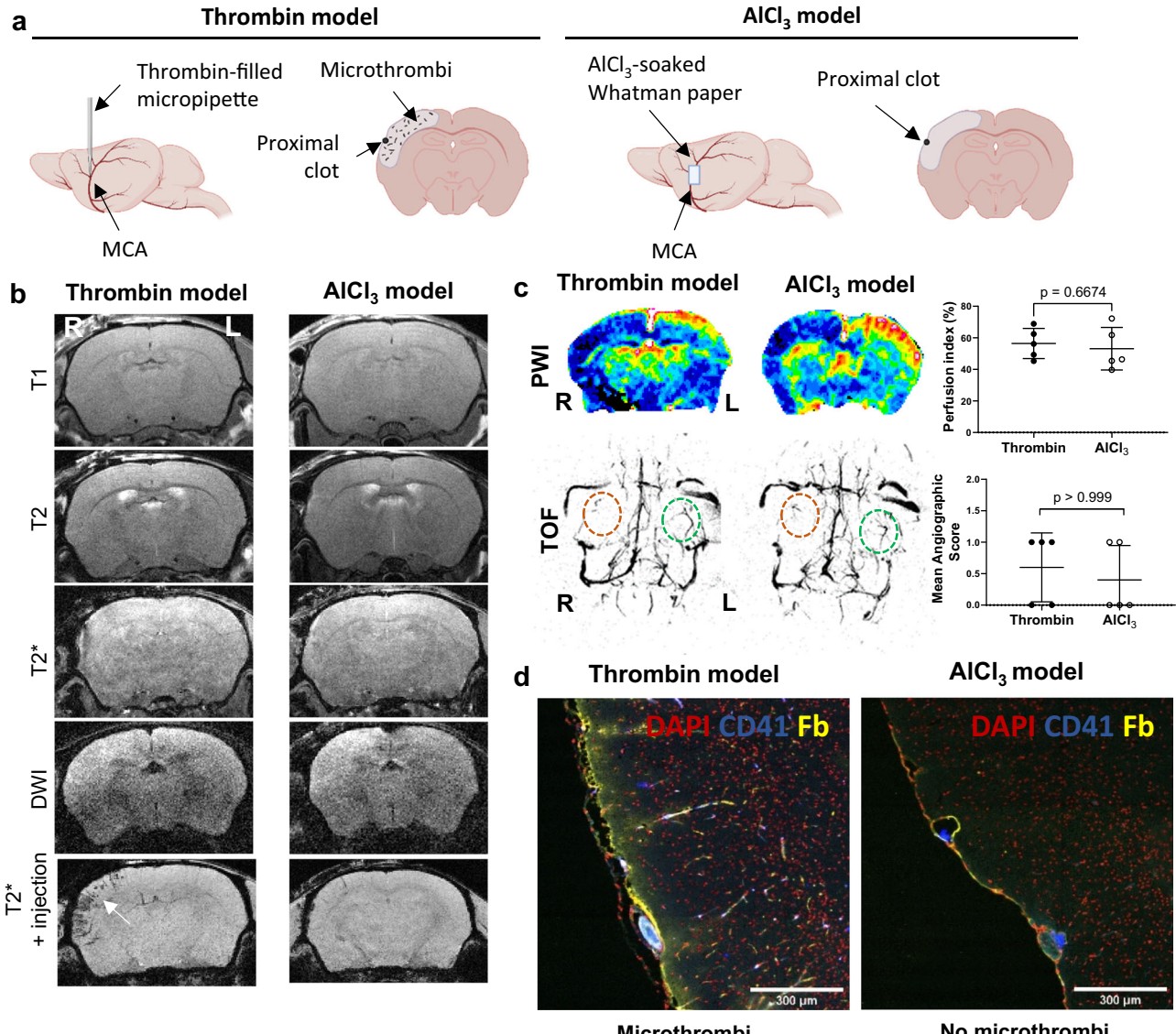

**Fig. 3 | Evaluation of MRI methods to detect differences between two MCAO models: thrombin model (with microthrombi) and AlCl3 model (without microthrombi). a** Description of the 'thrombin model' with thrombin injection in the MCA and 'AlCl3 model' with AlCl3 application on the MCA, and schematic coronal section of the brain in each model illustrating the presence or absence of microthrombi. Created with BioRender.com. **b** MRI images obtained using T1, T2, T2*-weighted, and diffusion-weighted imaging (DWI) after the induction of the MCAO models. Only the injection of PHySIOMIC in T2*-weighted imaging allows for the visualization of the microthrombi present in the thrombin model (indicated by the arrow). R and L indicate right and left side of the presented brain. **c** Study of cerebral perfusion using perfusion-weighted imaging (PWI, unpaired *t* test, two-sided) or angiography (TOF, Mann–Whitney, two-sided) in the models (*n* = 5 animals per group). Results are presented as mean ± SD. Source data are provided as a Source Data file. **d** Immunohistological images of the cerebral cortex were harvested 1 hour after the induction of each model. Microthrombi are identified through CD41 (platelet) and fibrin (yellow) staining, and nuclei are stained with DAPI (observations in 5 animals).

in particle biodistribution, immune system recognition, intracellular capture, and toxicity[32]. We hypothesized that the PC composition could facilitate passive interaction with microthrombi. To explore this aspect, we first engineered an artificial PC by incubating PHySIOMIC in a bovine serum albumin (BSA) buffer during the synthesis process, yielding the generation of saturated PHySIOMIC-BSA particles.

We then assessed, in vitro, the passive interaction of both PHySIOMIC and PHySIOMIC-BSA with microthrombi using a microfluidic system. Clots were generated within microfluidic chambers using human blood, and particles suspended in the blood were introduced into the system under a shear rate similar to that in arterioles. The formation of clots within the microfluidic chambers is represented by platelet clusters (stained with $DiOC_6$). The particles were observed via fluorescence microscopy, employing a reflection phenomenon that

enhances polydopamine due to its intrinsic light-absorbing characteristics (Fig. 5a). Although our experiments revealed a particle-to-microthrombi area ratio of 0.023 ± 0.005 in chambers perfused with PHySIOMIC, a lower ratio (0.003 ± 0.010, *n* = 4, *p* value = 0.0288) was observed in the chambers perfused with the PHySIOMIC functionalized with BSA (Fig. 5b). These findings were additionally validated by administering both particle types to mice following the induction of ischemic stroke. Indeed, $T_2$*-weighted scans disclosed that injecting PHySIOMIC-BSA did not yield any hypointensity signal, whereas injecting PHySIOMIC in the same animals resulted in a distinct signal void of 3.13 ± 2.3 mm³ (versus 0.56 ± 0.32 mm³, *n* = 5, *p* value = 0.0177), indicative of microthrombi presence (Fig. 5c, d).

We then analyzed the composition of the proteins forming the corona around our particles using mass spectrometry (Fig. 6a). Three

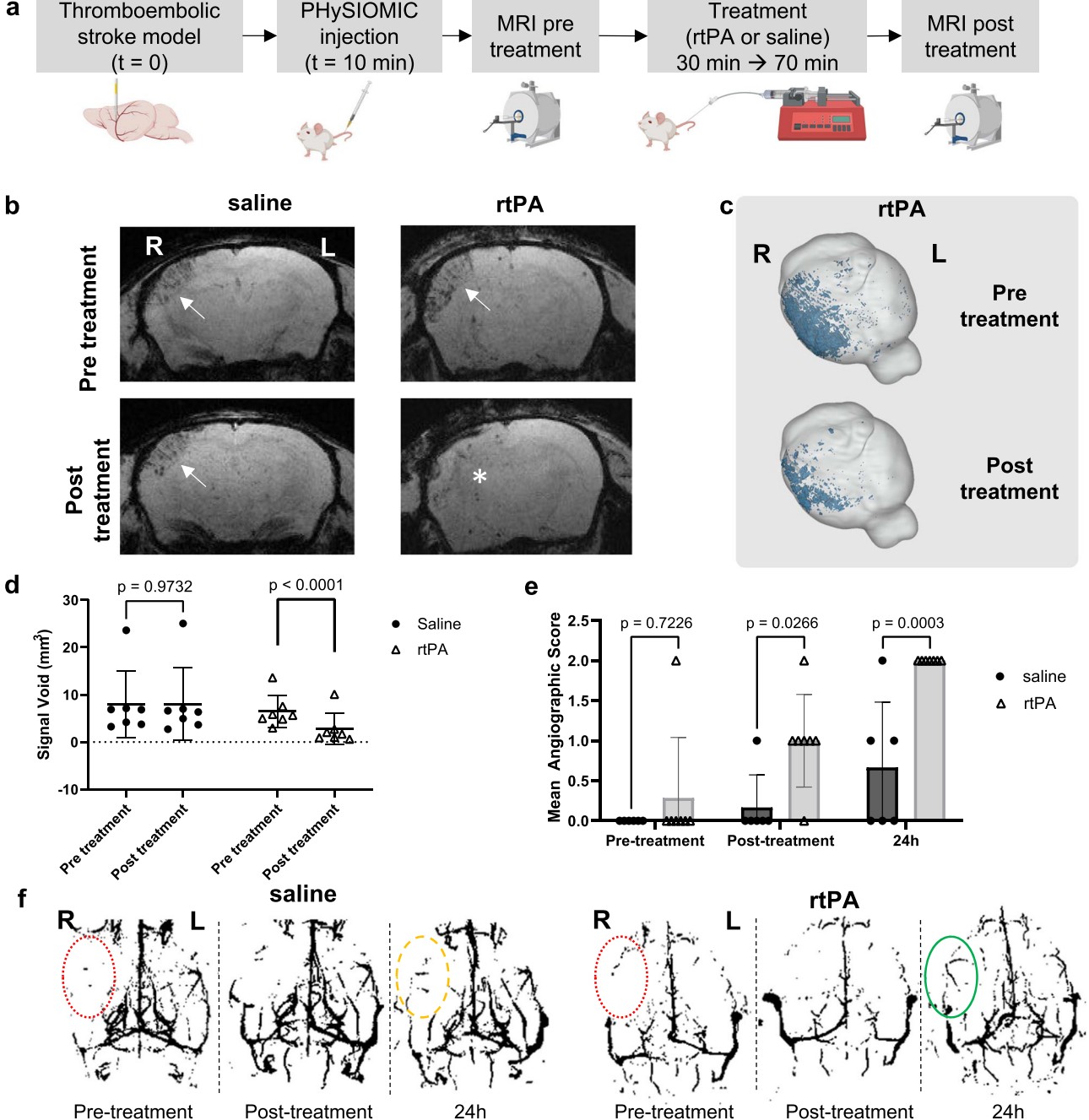

**Fig. 4 | PHySIOMIC reveals a significant reduction in the number of microthrombi after thrombolysis. a** Protocol illustration of PHySIOMIC injection 10 min after occlusion, followed by T2*-weighted sequence acquisition before and after treatment with either rtPA or saline. Created with BioRender.com.
**b**, **c** Representative T2*-weighted MRI scans and 3D reconstructed brain demonstrate microthrombosis in the ischemic cortex (arrows) and a significant decrease of the microthrombosis in the rtPA-treated group (asterisk). **d** Quantification of induced hyposignal reveals a diminution ($p < 0.0001$) in the treated group and no

changes ($p = 0.9732$) for the saline group ($n = 7$ animals, two-way ANOVA with Sidak's multiple comparisons test). **e**, **f** Magnetic resonance angiography displays no restoration of the blood flow in the MCA and downstream vasculature pre-treatment (red dotted lines), a partial restoration (yellow dashed line) for saline-treated mice, and a full restoration (green continuous line) for rtPA-treated mice ($n = 7$ animals, 2-way ANOVA). Data are presented as mean ± SD. R and L indicate right and left side of the presented brain. Source data are provided as a Source Data file.

forms of particles were examined i.e., PHySIOMIC, melamine particles (Fluorescent commercial microparticles with similar characteristics to PHySIOMIC) and carboxylated particles (melamine-COOH). Proteomics has revealed that the composition of the PC differs between PHySIOMIC and melamine particles when compared to melamine-COOH particles. This is particularly evident in the case of fibrinogen protein, which is found in higher quantities on melamine and PHySIOMIC particles (Fig. 6b). Additional information on the PC proteomic analysis can be found in Supplementary Fig. 9.

We conducted in vitro testing to assess the impact of PC on microthrombi targeting by observing melamine and melamine-COOH particles in a microfluidic system with exposition to blood or a phosphate buffer. The results demonstrated that the ratio of particles targeting microthrombi is higher for melamine particles when exposed to blood ($95.05 \pm 35.70$ mm$^{-2}$ for melamine compared to $24.26 \pm 10.31$ mm$^{-2}$ for melamine-COOH, $n = 4$, $p$ value = 0.0025) (Fig. 6c, d), thereby confirming the importance of the PC in the mechanism of passive targeting. The next microfluidic experiments

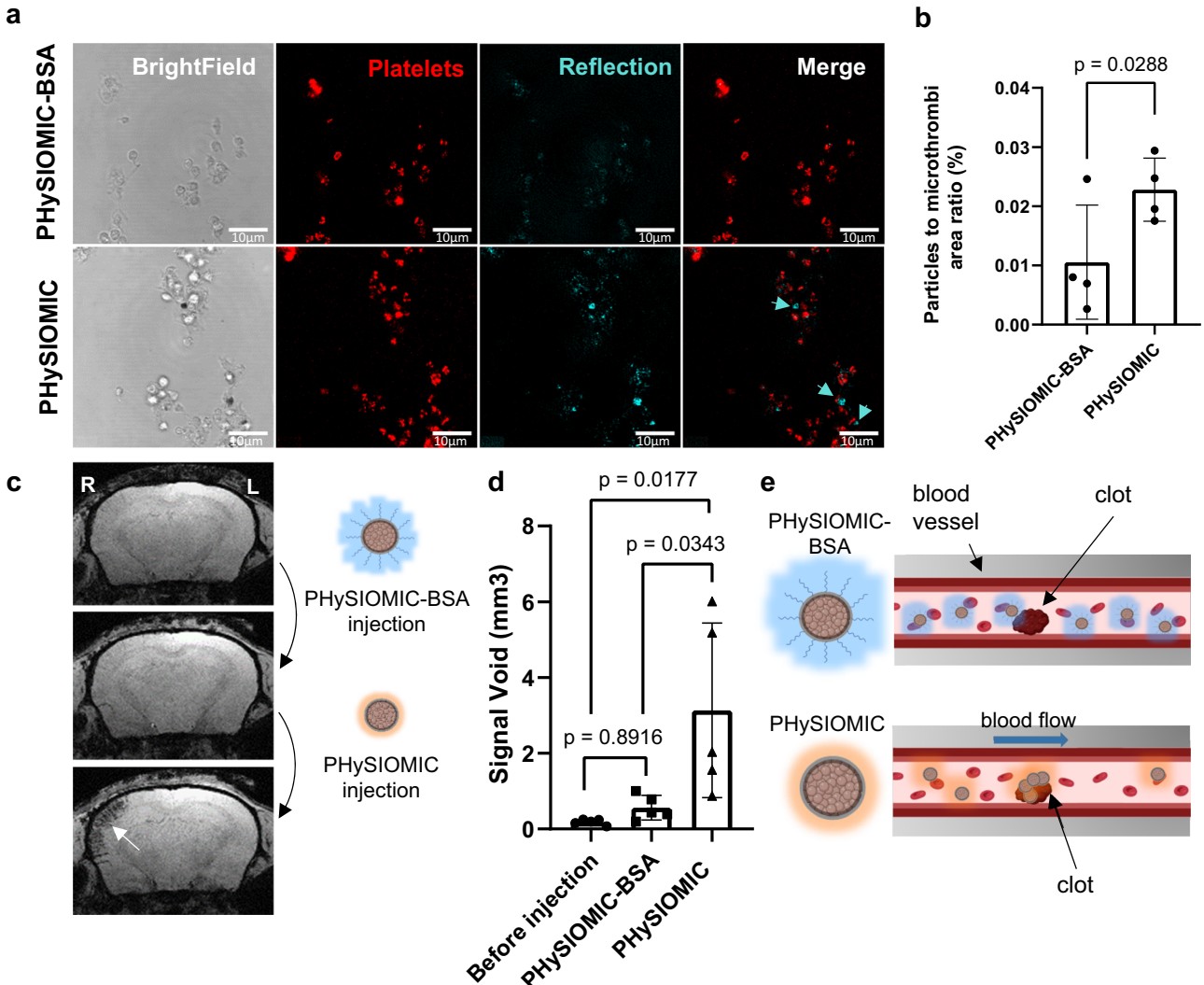

**Fig. 5 | PHySIOMIC failed to reveal microthrombi when saturated with albumin.**
**a** BSA-coated PHySIOMIC were designed to prevent protein corona formation in blood. Both PHySIOMIC and PHySIOMIC-BSA were injected in vitro, into a microfluidic chamber along with blood. Images were acquired through a combination of bright field and fluorescence microscopy, enhanced by a reflection acquisition technique to observe PHySIOMIC interacting with platelets-composed microthrombi. PHySIOMIC-BSA shows fewer occurrences in microthrombi compared to PHySIOMIC. Scale bar 10 μm. **b** Quantitative results presented in mean ± SD (*n* = 4 microfluidic chambers, paired *t* test, two-sided). **c**, **d** MRI scans of mice injected first with PHySIOMIC-BSA, then PHySIOMIC and subsequent quantification showing no decrease in signal with PHYSIOMIC-BSA, but a clear hyposignal after PHySIOMIC injection marking microthrombosis (*n* = 5 animals, One-way ANOVA with Tukey's multiple comparisons test). Results presented as mean ± SD. **e** Illustration of the proposed mechanism (Created with BioRender.com). PHySIOMIC-BSA forms a protein corona predominantly with BSA (in blue) and cannot target microthrombi, whereas PHySIOMIC develops a protein corona in vivo (in orange), enabling targeting to the clot. R and L indicate right and left side of the presented brain. Source data are provided as a Source Data file.

focused more specifically on the role of the fibrinogen in clot targeting. To this end, melamine particles were suspended in either fibrinogen-rich human plasma (Fg⁺) or fibrinogen-depleted plasma (Fb⁻) before injection into the system (Fig. 6e). Remarkably, we noticed an augmentation in the number of particles adhering to thrombi when the particles were suspended in fibrinogen-rich plasma (4.22 ± 3.36 in fibrinogen-rich plasma versus 1.38 ± 1.24 in fibrinogen-depleted plasma, *n* = 9, *p* value = 0.0078) (Fig. 6f). To measure the adsorption of fibrinogen onto the surface of particles, we exposed the particles to a fibrinogen solution (1 mg mL⁻¹) during 30 minutes at 37 °C. The particles were then centrifuged, and the fibrinogen remaining in the supernatant was measured using a nanospectrophotometer. Fibrinogen adsorption is greater on melamine and PHySIOMIC particles (50.0 ± 6.1 μg for Melamine, 72.1 ± 7.4 μg for PHySIOMIC, versus −4.08 ± 10.38 μg per 250 μl of particles for Melamine-COOH, *n* = 5) (Fig. 6g). Hence, the fibrinogen integrated into the PCs of PHySIOMIC plays a pivotal role in their passive targeting to thrombi and microthrombi (Fig. 6h).

Finally, we evaluated the in vivo targeting of our particles and the role of the fibrinogen present in the PC when injected in mice after ischemic stroke onset using two-photon microscopy. We observed distal microthrombi in the ischemic stroke model through a cranial window and examined the behavior of the particles. We noted a substantial accumulation of melamine particles at the periphery of microthrombi, as evidenced by platelet labeling with DiOC₆, while the number of melamine-COOH particles was minimal (Fig. 6h). This in vivo observation further confirms the role of fibrinogen, which is more abundant in the composition of the PC of melamine particles, in mediating the targeting of these particles to microthrombi.

**Biodistribution and biodegradation study of the PHySIOMIC**
Our next concern was to evaluate the correct elimination and biodegradation of PHySIOMIC in the body. We investigated the blood half-life of PHySIOMIC particles in mice using T₂*-weighted MRI scans. We monitored the particles' trajectory, particularly within the retro-orbital

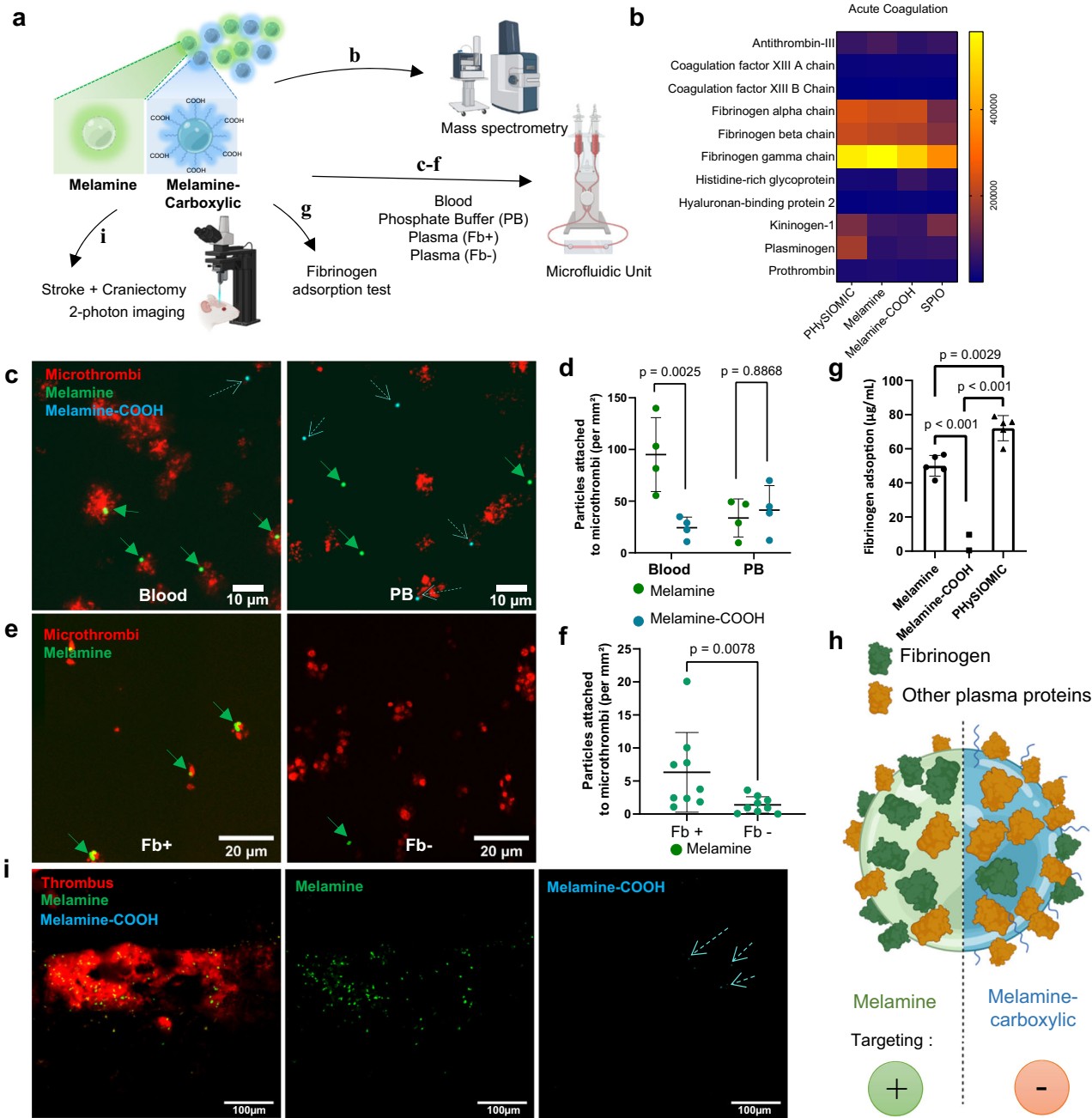

**Fig. 6 | Surface functionalization effects on protein corona composition and impact of fibrinogen on targeting of microthrombi. a** Protocol illustration for mass spectrometry, microfluidic experiments, and two-photon microscopy with commercial fluorescent melamine microparticles with no functionalization (Melamine, equivalent to PHySIOMIC) or with carboxylic termination (Melamine-COOH). Created with BioRender.com. **b** Proteomic analysis performed with PHySIOMIC, melamine and melamine-COOH. Proteins from the 'Acute coagulation' subgroup are presented (n = 3 pooled plasma samples). **c** Melamine (green arrows) and melamine-COOH particles (blue dashed arrows) were injected either with Phosphate Buffer (PB) (no formation of protein corona) or whole blood (protein corona formation) in a microfluidic system where microthrombi were previously formed. **d** Quantification of the number of particles targeting microthrombi (expressed in the number per microthrombi surface). A statistical difference is found between melamine and melamine-COOH particles only when injected with blood. Results are expressed as mean ± SD (n = 4 microfluidic chambers, Student's t test, two-

sided). **e** Melamine particles were injected into the microfluidic system with either normal plasma (Fb+) or fibrinogen-depleted plasma (Fb−). **f** Quantification of the targeting of particles to microthrombi is higher in normal plasma than in fibrinogen-depleted plasma. Results are expressed as mean ± SD (n = 9 microfluidic chambers, Mann–Whitney test, two-sided). **g** Fibrinogen adsorption on the surface on melamine, melamine-COOH and PHySIOMIC particles presented as mean ± SD (n = 5 replicated sample measurements, One-way ANOVA with Tukey's multiple comparisons), after a 30 minutes exposition to a fibrinogen solution at 37 °C. **h** Diagram illustrating the potential impact of fibrinogen on protein corona composition in relation to microthrombi passive targeting. Created with BioRender.com. **i** Two-photon observations of melamine and melamine-COOH particles injected intravenously into a mouse after a thrombin-induced stroke, showing the specific melamine attachment around thrombi in vivo (n = 3 animals). Source data are provided as a Source Data file.

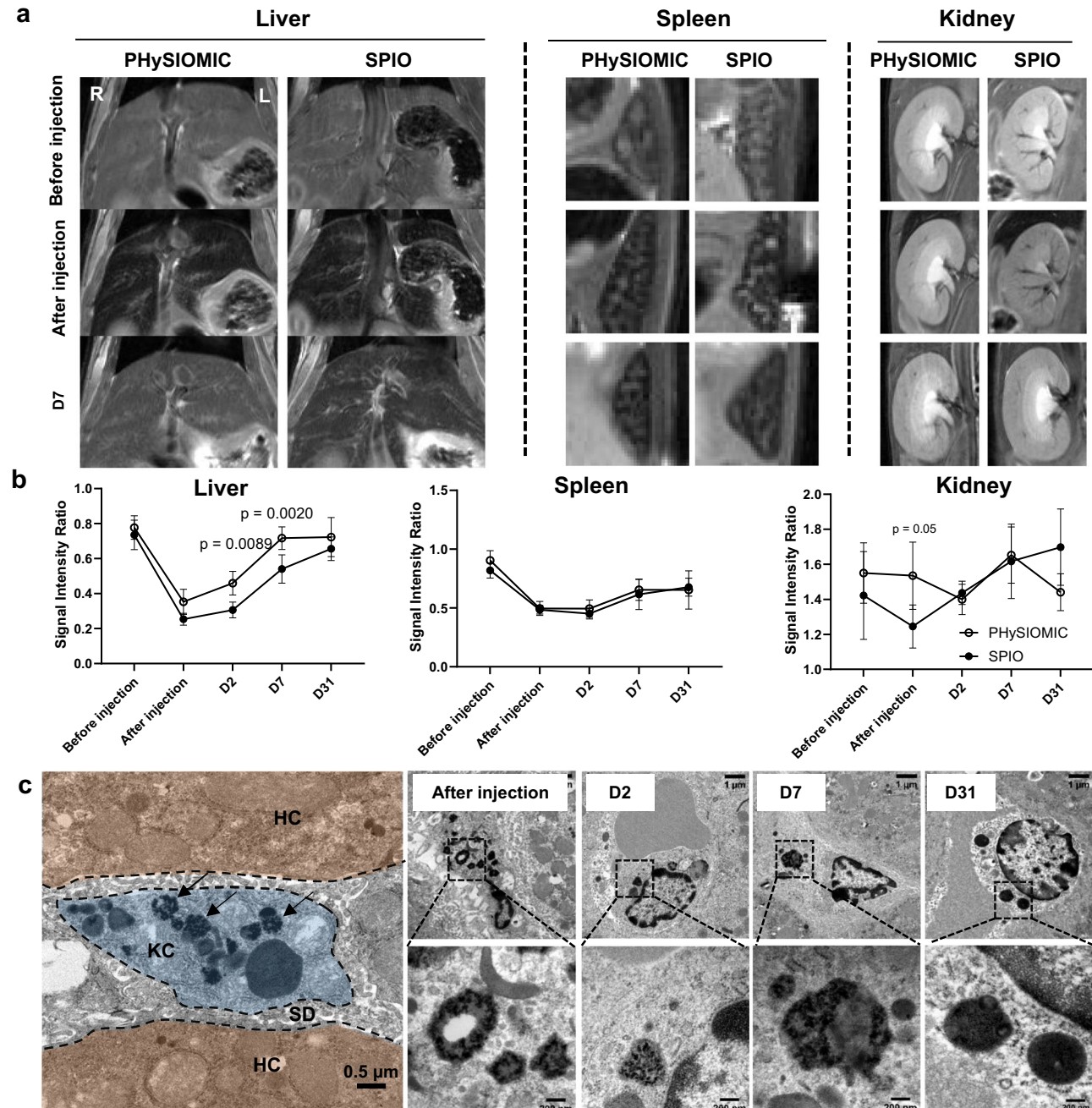

**Fig. 7 | Similar biodistribution and biodegradation of PHySIOMIC and SPIO in liver and spleen, evidenced by TEM and MRI. a** Longitudinal T2-weighted images acquired before and after injection of PHySIOMIC and SPIO (4 mg kg⁻¹), and at 2 and 7 days post injection. **b** Quantification of T2 intensity values in the liver, spleen, and kidneys. Faster biodegradation is observed in the Liver with PHySIOMIC, and a part of injected SPIO is eliminated through the kidney. Results are expressed as mean ± SD ($n$ = 5 animals, Two-way ANOVA with Sidak's multiple comparisons test). **c** TEM images of the degradation of PHySIOMIC in the Kupffer cells (KC, blue) located inside the Space of Diss (SD, gray) between the hepatocytes (HC, brown) of the liver. PHySIOMIC (arrows) are observed in lysosomes of the Kupffer cells where the structure of clusters observed within lysosomes undergoes significant alteration after one month ($n$ = 2 animals). Source data are provided as a Source Data file.

vein, and quantified the signal reduction as they traversed. To provide a basis for comparison, the same experiment was replicated using SPIO particles (Supplementary Fig. 10a). While the signal caused by SPIO injection exhibited consistency over time (spanning over 2 minutes), we noted a gradual restoration of the signal to its baseline level for PHySIOMIC particles (Supplementary Fig. 10b). This observation corresponds with prior research findings that highlight a notably brief blood half-life for clusters such as PHySIOMIC compared to the significantly longer half-life observed for nanoparticles such as SPIO[21].

To ensure the effective biodegradation of PHySIOMIC, facilitated by the biodegradable polydopamine, we injected a high concentration

of particles, equivalent to 4 mg of iron per kg, to healthy mice and compared it with the SPIO, which are known to be biodegradable. Whole-body MRI was performed before and shortly after the injection of PHySIOMIC or SPIO suspensions. Signal enhancement was monitored in the liver, spleen, and kidney at 2, 7, and 31 days after injection. Both PHySIOMIC and SPIO particles induced a strong hyposignal within the liver and the spleen (Fig. 7a), which is in line with a rapid elimination of particles from the bloodstream and uptake by the mononuclear phagocyte system; in the Kupffer cells of the liver and resident macrophages of the spleen[21]. This hyposignal gradually fades out in time ending at baseline level at 31 days post injection (Signal

Void of $1.55 \pm 0.17\,mm^3$ before injection versus $1.44 \pm 0.10\,mm^3$ at 31 days, $n = 5$, $p$ value = 0.6323). This observation was attributed to the gradual biodegradation of the iron oxide particles within the lysosomal compartment of the macrophages; a mechanism well described in literature for inorganic nanoparticles, including iron oxide[33]. There is a residual hyposignal after 31 days post-injection in the spleen for both types of particles that may be explained by the degradation of iron oxide into iron-rich ferritin proteins, mainly accumulated in the spleen[20]. Following the administration of SPIO particles, a hypointense signal is noticeable in the kidneys, whereas no alteration is observed with PHySIOMIC (Fig. 7b). PHySIOMIC presenting a larger hydro-dynamic diameter does not pass through the renal clearance barrier for elimination[34]. This implies that PHySIOMIC particles would not be contraindicated for patients with kidney impairments, as was observed with certain SPIO nanoparticles.

To further investigate the biotransformation of PHySIOMIC within the liver, mouse liver samples were collected at the same time points as the MRI scans and subjected to TEM observation (Fig. 7c). As anticipated, PHySIOMIC particles were identified within the Kupffer cells of the liver post-injection, and their biotransformation was observable within the lysosomes of these cells. Over time, a degradation of the particle structure within the lysosomes can be observed, consistent with the loss of superparamagnetic properties observed in MRI. Taken together, the results of the biodegradation study indicate a breakdown of the polydopamine matrix due to the low pH and the presence of enzymes in the lysosomes[21] and subsequent elimination of iron oxide into iron-rich ferritin protein, contributing to the restoration of the body's iron pool.

## Discussion

We have developed a contrast agent called PHySIOMIC to be revealed by MRI, synthesized through a self-assembly process of SPIO within a polydopamine matrix. This approach has enabled us to combine the advantages of clinically approved SPIO, which possesses good $T_2^*$ relaxivity in MRI, with those of micro-sized particles. These micro-sized particles enhance $T_2^*$ contrast and enable rapid absorption for fast molecular imaging. Additionally, the polydopamine matrix conferred thrombosis-targeting properties that enabled molecular MRI of microthrombi in a thromboembolic ischemic stroke murine model. It is also interesting to note that the PHySIOMIC provided strong imaging signal detection, although they exhibited a short half-life (less than a minute) compared to the SPIO, which typically circulates much longer (over 10 minutes) but was not able to reveal the microthrombi. This might appear surprising as the field of nanomedicine usually supports long circulation half-life as a positive feature for molecular imaging applications as longer exposure time should logically result in increased targeting[35]. However, this study provides evidence that this reasoning is not always true and that, at least in some contexts, modifying the surface to improve the targeting properties can be more valuable than increasing the circulation time.

Due to the light-absorbing properties of polydopamine, it was not possible to obtain fluorescently labeled polydopamine particles without surface modification for in vitro experiments. Therefore, to delve deeper into the thrombosis targeting mechanism identified with PHySIOMIC particles, we opted to use commercially available fluorescent particles composed of melamine resin. This choice was motivated by the presence of amine groups on the surface of melamine resin, which is akin to polydopamine. Furthermore, it has been reported that modifying the external layer of nanomedicines with carboxylate groups significantly influences the profile of adsorbed proteins compared to bare particles with amine groups[36]. Consequently, we also used melamine functionalized with carboxylic termination (melamine-COOH). Thanks to the strong fluorescent labeling of melamine and melamine-COOH, we were able to track them in vitro through a microfluidic study and in vivo with two-photon microscopy.

While PC impact on nanoparticles is well-documented in the literature[37], only a restricted number of studies have deeply investigated the potential attributes the PC might possess for targeting clinical objectives. It is noteworthy that the PC plays a significant role in biodistribution, blood circulation time, and immune response to particles. Thus, the presence of opsonins within this PC facilitates rapid recognition by the complement system and the clearance of particles from circulation by macrophages. Our study has showcased, through PHySIOMIC particle saturation with BSA and subsequent microfluidic experiments with melamine in diverse fluids, that the PHySIOMIC-microthrombi interaction process is contingent upon the formation of a PC on the particle surface. Thus, we notably investigated the composition of the PC of PHySIOMIC particles and pursued the hypothesis that its fibrinogen composition could contribute to the targeting process of microthrombi. However, it remains plausible that other PC components might contribute to the thrombosis-targeting phenomenon. In such conditions, it should be noted that the presence of a higher concentration of fibrinogen in the PC on the PHySIOMIC particles could also have prothrombotic consequences. At this stage, we did not completely elucidate to which specific thrombosis component or protein the PC confer adhesion, but one possible target could be the activated form of platelet integrin αIIbβ3, which is known to specifically recognize fibrinogen[38]. This hypothesis is in accordance with the TEM observations, where the PHySIOMIC was found attached to activated platelets. So, in addition to bringing some fibrinogen to the edge of the clot and thus possibly enhancing fibrin polymerization, the PHySIOMIC covered with fibrinogen could also increase platelet aggregation. Besides, although the clot lysis assay did not show a significant difference in the tPA-induced clot lysis time in the presence of PHySIOMIC, the mean value of lysis time was numerically higher. Overall, the putative risk of prothrombotic effect should be monitored for further developments of the PHySIOMIC particles.

Another risk to consider is the potential toxicity induced by PHySIOMIC particles that would eventually cross into the brain tissue, especially in the context of severe hemorrhagic transformation post-stroke. Although none of the current data did reveal PHySIOMIC retention other than the one bound to thrombi (due to their sub-micrometric size that prevents their passage through the blood-brain barrier (BBB)), the fate of the PHySIOMIC would need to be closely monitored for future development. Notably, a clear asset of this contrast agent is that it is fully made from biocompatible materials (iron oxide and polydopamine) that lack neurotoxic side effects. Accordingly, SPIO that can cross the injured BBB has been injected in humans at the subacute phase of ischemic stroke, without worrying signals for toxicity[39]. Moreover, polydopamine nanoparticles exhibit strong anti-oxidant properties and were shown to result in a neuroprotective effect in an ischemic stroke model for this specific reason[40]. These data are reinsuring for the clinical translation of PHySIOMIC.

The presence of distal microthrombi can often remain imperceptible in ischemic stroke models, mirroring the situation in stroke patients. This is problematic as it influences treatment response and potential complications. In vivo injection of PHySIOMIC in a thrombin-induced stroke mouse model reveals microthrombosis in $T_2^*$-weighted MRI scans at the acute stage of stroke. The monitoring of microthrombosis over time also exhibited consistency with the spontaneous reperfusion that occurs in this specific model initiating from 6 hours after occlusion. Comparisons were made with an alternate model of ischemic stroke where thrombosis formation is induced in situ via the topical application of aluminum chloride ($AlCl_3$) in the MCA[31]. Interestingly, MRI sequences classically performed for stroke patients failed to reveal any differences between the two models. Although PHySIOMIC unveiled the microthrombi existing within the intra-luminal thrombin injection model, it failed in the $AlCl_3$ model. Hence, PHySIOMIC offers a method to visualize both the presence and absence of microthrombi, as well as quantify their prevalence. This approach has

the potential to enhance clinical management by providing a more precise assessment of thrombotic conditions in patients with ischemic stroke.

In addition, the injection of PHySIOMIC particles into various animal models, both in stroke and non-stroke conditions (transient ischemic attack and necrosis model), has uncovered the presence of thrombosis (Supplementary Fig. 11). We can postulate that PHySIOMIC has the potential to enhance the characterization of stroke etiology by identifying the presence of microthrombi, whether occurring at the acute phase or as a secondary response to ischemia. Monitoring microthrombi could also enable the testing of various therapeutic agents specifically designed to target microthrombi in stroke patients, including those undergoing recanalization through thrombectomy. Numerous studies are currently exploring the development of thrombolytic agents designed to transport treatment directly to the clot[41,42], thereby reducing the risk of hemorrhage and potentially broadening the scope of individuals who can benefit from treatment[43]. Conversely, the innate ability of PHySIOMIC for passive targeting of microthrombi, combined with a high surface loading capacity, make them an attractive carrier for delivering thrombolytic agents.

Functionalized nanoparticles with specific fibrin-binding ligands have been previously crafted[44,45], holding the potential for precise thrombus imaging. However, these nanoparticles come with the drawback of yielding a rather faint MRI signal. In response to this challenge, microparticles of iron oxide have been synthesized, exhibiting a stronger MRI signal to observe thrombi[46]. Yet, this approach failed to reveal microthrombosis in the brain or the thrombosis involved in ischemic stroke. Moreover, they face the issue of lacking biocompatibility due to their coating composed of polyurethane and polystyrene, leading to accumulation in specific liver macrophages, Kupffer cells[47]. In our study, we have shown that the PHySIOMIC is degraded within the liver Kupffer cells, similarly to FDA-approved SPIO nanoparticles. This aspect will enable its future clinical development, although further safety studies are still necessary.

In conclusion, we have identified a approach to target thrombosis in vivo exploiting the PC formation. We exploited this targeting pathway to develop a biocompatible contrast agent with the potential to unveil and monitor microthrombosis in stroke patients, providing valuable insights for guiding treatment strategies.

## Methods

### PHySIOMIC synthesis and functionalization

PHySIOMIC was synthesized in a self-assembly method by mixing SPIO and dopamine. SPIO particles (Vivotrax, Magnetic Insight Inc.) are composed of iron oxide with a dextran coating, the same composition as the Ferucarbotran (Resovist) contrast agent, which was clinically authorized until 2009. Briefly, 200 μL of SPIO nanoparticles solution (5.5 mg mL$^{-1}$) were mixed with dopamine (10 mg mL$^{-1}$, Sigma-Aldrich) in a TRIS buffer (10 mM, pH 8.8). A reaction of polymerization of dopamine into polydopamine occurs under constant agitation at room temperature for 24 h, as described elsewhere[48]. The polymerization reaction is stopped by washing particles in a phosphate buffer (10 mM, pH 8.8) using a separating magnet (PolyATract® System 1000 Magnetic Separation Stand, Promega). The solution is then placed under sonication (Ultrasonic Probe Sonicator, Hielscher®) at high intensity for 5 minutes to obtain clusters of the requested size.

PHySIOMIC was functionalized with BSA to saturate the PC. PHySIOMIC is washed in a phosphate buffer 0.1 M with BSA (10 mg mL$^{-1}$, Sigma-Aldrich), and the reaction is left for 24 h, under constant agitation at 37 °C. The particles are washed and resuspended in a Phosphate Buffer (PB 10 mM). For in vivo injection, clusters were washed and resuspended in a mannitol buffer (0.3 M, pH 7.4). All particle suspensions are kept under agitation at 4 °C until used.

Commercial microparticles of 1 μm based on melamine resin (Sigma-Aldrich) with amine (Rhodamine B-marked) or carboxylic terminations (Nile Blue-marked) are prepared for the microfluidic and two-photon microscopy experiment at respective volume ratios of 1:100 and 1:1000 in mannitol buffer.

### Size and zeta potential measurement

DLS measurements were carried out on a Zeta sizer Ultra (Malvern Panalytical®) equipped with a 633 nm laser at a fixed scattering angle of 173°. Particle suspension was diluted 1/50 (v/v) in 10 mM PB, to ensure the appropriate scattering intensity in the detector before measurements. The average hydrodynamic diameter (HD) and the PDI measurements are determined at 25 °C. Zeta potential measurements are acquired after the sample dispersion in $10^{-3}$ M KCl using the same equipment with a DTS 1070 cell at 25 °C, with a dielectric constant of 78.5, a refractive index of 1.33, a viscosity of 0.8872 cP, and a cell voltage of 150 V. The zeta potential was calculated from the electrophoretic mobility using the Smoluchowski equation. All measurements were performed in triplicate.

### Iron quantification with ferrozine assay

Total iron is quantified using a modified version of the ferrozine colorimetric assay[49]. A 500 μL of 2 N HCl is added to 500 μL of sample lysate. The iron standards are prepared using increasing concentrations of FeCl$_2$. Samples are incubated overnight and mixed with iron detection reagent (37.5 μL of 5 mM ferrozine, 60 μL of ammonium acetate 30% and 30 μL of ascorbic acid 30%; Sigma-Aldrich). Equal volumes of the test and standard samples were deposited into a 96-well microplate in duplicates and absorbance is read at 560 nm using a microplate reader (ELx808 Absorbance Reader, Biotek Instruments).

### Transmission electron microscopy

For observation of SPIO or PHySIOMIC in suspension, a droplet of particles was deposited and dried on a glow-discharged (hydrophilized) 200-mesh formvar/carbon grid. To observe liver sections, small samples were harvested at different times after injection and fixed in a 2.5% glutaraldehyde solution in 0.1 M cacodylate buffer (EMS), rinsed with cacodylate 0.1 M, post-fixed with osmium tetroxide in 0,1 M cacodylate and rinsed with cacodylate 0.1 M. The samples were then dehydrated in progressive bath of ethanol (70–100%), finally replaced by acetone, and embedded in an Embed 812 resin mixture. After 48 hours of polymerization at 60 °C ultrathin sections of 80 nm were obtained using an ultramicrotome (Leica ultracut R). Images of SPIO, PHySIOMIC, and liver sections were captured using an ORIUS 200 digital camera (GATAN) mounted on a transmission electron microscope 1011 (JEOL). Samples treatment and image acquisition were performed at CMAbio3 (Center de Microscopie Appliquée à la Biologie, Biological Microscopy Facility Center UFR des Sciences, US EMerode, University of Caen Normandy). Microthrombi in brain section were observed at the electron microscopy imaging facility MicroEleCS (INSERM UMR-S1255 Unit, Strasbourg, France). Brain tissue was fixed with 2.5% glutaraldehyde in 0.1 M sodium cacodylate buffer containing 2% sucrose, previously warmed to 37 °C (305 mOsm, pH 7.3) for 1 hour. The samples were then rinsed and post-fixed with 1% osmium tetroxide in 0.1 M sodium cacodylate buffer for 1 hour at 4 °C. After additional washing in the 0.1 M sodium cacodylate buffer, the samples were dehydrated in successively increasing ethanol concentrations before embedding in epoxy. The resin was polymerized for 2 days at 50 °C. Ultrathin sections (100 nm) were stained with lead citrate and uranyl acetate, and examined under a Jeol 2100-plus (120 kV).

### Relaxivity measurements

Samples of PHySIOMIC and SPIO at increasing concentrations (0, 0.72, 1.43, 2.15, 2.86, 3.58 mM) were prepared. The samples were diluted in a TAE buffer (40 mM Tris-acetate, 1 mM EDTA) with 2% agarose after heating the solution until boiling. The mixture was then placed in 500 μL Eppendorf tubes. The device containing the particle samples was positioned at the center of a BioSpec 7-T TEP-MRI (Bruker,

Germany). r1 relaxivity was calculated from $T_1$ mapping using the flow-sensitive alternating inversion recovery–rapid acquisition with relaxation enhancement (FAIR-RARE) sequence with a repetition time (TR) of 3000 ms and an inversion time ranging from 6.5 to 2000 ms. r2 relaxivity was obtained from $T_2$ mapping using a multislice multiecho (MSME) sequence with a TR of 4000 ms and an echo time (TE) ranging from 3.65 to 51.11 ms, while r2* relaxivity was derived from $T_2$ mapping using a multigradient echo sequence with a TR of 4000 ms and a TE ranging from 2 to 17.47 ms. Relaxivity values were calculated as $R = R0 + r \times [Fe]$, with R as the relaxation rate (1/T), R0 as the relaxation rate of the solution without particles, [Fe] as the iron concentration of particles and, finally, r as the relaxivity value.

## Microfluidic experiment

The microfluidic experiment was conducted using an Ibidi® pump system to control the blood flow through a microfluidic channel (μ-slide VI 0.4, Ibidi®) to reproduce thrombosis in vitro. First, the microfluidic channel was incubated with collagen from calf skin (Sigma) and left at room temperature for an hour. Whole blood collected on citrate from healthy volunteers was provided by the local blood bank, the Etablissement Français du Sang (EFS, Caen, France), following the guidelines from the "agreement for the transfer of products derived from blood or its components for non-therapeutic purposes n° PLER /2021/005" established between the EFS Hauts-de-France−Normandie and the Institut National de la Santé et de la Recherche Médical (INSERM). This agreement ensures good practice for venipuncture and the use of blood from healthy volunteers in research, including the inclusion of informed consent obtained from participants. This agreement has not been reviewed by an institutional review board as the whole activity of the EFS comply to the ethical requirement. Platelets of collected blood were stained using $DiOC_6$ intracellular probe (1 μg mL$^{-1}$, Abcam). Blood was perfused through the microfluidic channel in contact with collagen, reproducing an endothelial damage in the vasculature, at a shear rate of 500 s$^{-1}$ according to previous study[50]. The collagen coating induced platelet aggregation, and the formation of microthrombi that could be observed in the AF488 channel with a confocal microscope (Leica). Plasma was obtained from whole blood tube samples after two steps of centrifugation (15 min at 200 g). Fibrinogen-depleted plasma was obtained from Cryopep (Montpellier, France). After extended washing of the blood in the chamber with physiological serum, 200 μL of a preparation of particles (PHySIOMIC or melamine microparticles) suspended in either 2 mL of PB, blood, plasma (Fb+), or plasma depleted in fibrinogen (Fb−) were injected into the chamber and observed in the corresponding fluorescence channel. Images were acquired using Leica Aplication Suite software (LAS AF, v.3).

## PC isolation and proteomic study

PHySIOMIC, melamine-$NH_2$ and melamine-COOH were incubated in human plasma for 1 hour at 37 °C to induce the formation of a PC on their surface. The particles were washed three times through centrifugation (2000 × g during 5 minutes) and resuspended in PB (10 mM, pH 8.5). Particles are then resuspended in a lysis buffer (RIPA Buffer, ThermoFisher) to separate PC from particles. A centrifugation (10,000 × g for 5 minutes) is made to obtain the particles in the pellets (without PC) and the supernatant (lysis buffer and proteins) is sent for proteomic analysis, at Proteogen Platform (University of Caen Normandy). Five μg of each protein extract were prepared using a modified Gel-aided Sample Preparation protocol[51]. Samples were digested with trypsin/Lys-C overnight at 37 °C. For nano-LC fragmentation, protein or peptide samples were first desalted and concentrated onto a μC18 Omix (Agilent) before analysis. The chromatography step was performed on a NanoElute (Bruker Daltonics) ultra-high-pressure nano flow chromatography system. Approximately 200 ng of each peptide sample were concentrated onto a C18 pepmap 100 (5 mm × 300 μm

i.d.) precolumn (Thermo Scientific) and separated at 50 °C onto a reversed-phase Reprosil column (25 cm × 75 μm i.d.) packed with 1.6 μm C18 coated porous silica beads (Ionopticks). Mobile phases consisted of 0.1% formic acid, 99.9% water (v/v) (A) and 0.1% formic acid in 99.9% ACN (v/v) (B). The nanoflow rate was set at 300 nl/min, and the gradient profile was as follows: from 2 to 15% B within 15 min, followed by an increase to 25% B within 10 min and to 37% B within 12 min and further to 9% within 7 min and reequilibration. Mass spectrometry (MS) experiments were carried out on an TIMS-TOF pro mass spectrometer (Bruker Daltonics) with a modified nano electrospray ion source (CaptiveSpray, Bruker Daltonics). A 1400 spray voltage with a capillary temperature of 180 °C was typically employed for ionizing. MS spectra were acquired in the positive mode in the mass range from 100 to 1700 m/z and 0.75 to 1.30 1/k0 window. In the experiments described here, the mass spectrometer was operated in PASEF DIA mode with exclusion of single-charged peptides. The DIA acquisition scheme consisted of 24 variable windows ranging from 300 to 1000 m/z. More details on the proteomic data generated in this study are provided in the Supplementary Information (Supplementary Fig. 9), and in Source Data file. The data are available in the iProX database under accession code IPX0008212001. Proteins were classed in terms of biological function, as seen in a previous publication[52].

## Fibrinogen adsorption quantification

Melamine (diluted at 1:5), Melamine-COOH (diluted at 1:50) and PHySIOMIC particles were incubated with a fibrinogen solution (1 mg mL$^{-1}$) at a 1:1 ratio during 30 minutes at 37 °C. The particles were then centrifuged (3 minutes, 10,000 × g) and the supernatant was measured with a nanospectrophotometer (Nanodrop, Implen) with a molar extinction coefficient of 1,51. The amount of fibrinogen adsorbed onto the surface of the particles is measured by subtracting the fibrinogen value measured in the supernatant from the control fibrinogen value (incubation of fibrinogen and mannitol buffer of the particles).

## Animals

All experiments involving animals were conducted in accordance with European Council (Directives of November 24, 1986 (86/609/EEC) and French Legislation (act no, 87-848) on Animal Experimentation and approved by the local ethical committee of Normandy (CENOMEXA – comité d'éthique normand en matière d'expérimentation animale, APAFIS #13172). All studies were conducted on male or female Swiss mice (age 8–10 weeks; weight 35–45 g; CURB, Caen, France). Mice were housed in a temperature-controlled room (21 °C), hygrometry (55%), brightness (100 lux), on a 12-hour light/12-hour dark cycle with food and water ad libitum. They were accommodated in small stable social groups (3 to 5 per cage) with a minimum enrichment (a cardboard roll + a sheet of paper towel), and handled only by experienced staff. Before surgery, mice were deeply anesthetized with isoflurane 5% in a 70/30 gas mixture (NO2/O2) and maintained under anesthesia with 2% isoflurane in a 50/50 gas mixture ($NO_2/O_2$). To avoid any pain due to surgery, mice were injected subcutaneously with buprenorphine (Buprecare®, 0.1 mg kg$^{-1}$), 20 minutes before. The animals' temperature was maintained at 37 ± 0.5 °C throughout the surgical procedures using a feedback-regulated heating system with a rectal probe. A catheter was inserted into the tail vein of mice for intravenous administration of contrast agent or treatments. After surgery, animals were allowed to recover in a clean heated cage.

## MCAO models

As described in Orset et al.[28], mice were placed in a stereotaxic device, the skin between the right eye and the right ear was incised, and the temporal muscle was retracted. We performed a small craniotomy followed by the excision of the dura, to expose the MCA. A home-made glass micropipette was introduced into the lumen of the MCA, and 1 μL of purified murine alpha-thrombin (1 UE; Stago BNL) was

pneumatically injected to induce the in situ formation of the clots. The pipette is removed 10 minutes after the injection at which time the clot was stabilized. For the aluminum chloride model (AlCl₃), the MCA was exposed, and AlCl₃ (Sigma-Aldrich) was topically applied on the artery (as previously described[53]). To study the effects of thrombolysis on microthrombi, mice received intravenous administration of rtPA (10 mg kg⁻¹ in 200 μL; Actilyse) as 10% bolus and 90% perfusion over 40 minutes. The control group received the same volume of saline under the same conditions.

### Intra-striatal necrosis model
Mice were positioned in a stereotaxic frame, and an incision was made in the skin on top of the head. A craniotomy was performed at specific coordinates relative to bregma: −2 mm in the left-right axis and −0.5 mm on the antero-posterior axis. A custom-made glass micropipette was inserted into the brain at a depth of −3 mm from the brain's surface. Subsequently, 1 μL of staurosporine (2 mg mL⁻¹, Fisher Scientific) was injected into the striatum to induce neuronal cell death[54]. The micropipette was then withdrawn, and the skin was sutured.

### AIT model
In this previously described model[55], the MCA bifurcation was exposed following a small craniectomy, while keeping the dura intact. A micropipette was positioned at the bifurcation until achieving a reduction of blood flow (<80%), which was monitored using laser Doppler flowmetry (Oxford Optronix). The pressure was maintained for 15 minutes, after which the pipette was removed, allowing the restoration of blood flow.

### Exclusion criteria
Mice were excluded in case of death during the experimental procedure, technical problems, or hemorrhage, or if mice presented a lesion size at 24 h post-occlusion superior to 35 mm² or inferior to 10 mm² (without injection of a thrombolytic agent).

### Magnetic resonance imaging (MRI)
In vivo experiments on stroke imaging were carried out on a Pharmascan 7 T/16 cm system with surface coils (Brüker, Germany), and acquired with Paravision Software (v6.0.1, Bruker). Magnetic resonance angiographies (MRA) were performed using a 2D-TOF sequence (TE/TR 5/12 ms). A $T_1$-weighted rapid acquisition with relaxation enhancement (RARE) sequence was used to obtain anatomical images (TE/TR 9/1200 ms, with $70 \times 70 \times 500\ \mu m^3$ spatial resolution. $T_2$-weighted images were acquired using a multislice multi-echo (MSME) sequence (TE/TR 50/3000 ms with $70 \times 70 \times 500\ \mu m^3$ spatial resolution. 3D $T_2$*-weighted gradient-echo imaging with flow compensation (GEFC, spatial resolution of $93 \times 70 \times 70\ \mu m$ interpolated to an isotropic resolution of 70 μm) with TE/TR 9/50 ms and a flip angle of 15° was performed to visualize PHySIOMIC. Diffusion-weighted images (DWI) were acquired using a standard spin echo imaging modified with a Stejskal-Tanner gradient scheme (TE/TR 38 ms/2000 ms, with $75 \times 75 \times 500\ \mu m^3$ spatial resolution, giving an in-plane resolution of 100 x 78 mm, slice thickness of 0.75 mm, one direction diffusion gradient, in the frequency encoding direction) with a b factor of 1000 s/mm² and one averaging. For perfusion-weighted imaging, a gradient-echo fast imaging with steady-state precession sequence was used, with TR/TE, 10/5 ms (FA of 8°), temporal resolution of 911 ms, and spatial resolution of $100 \times 100 \times 75\ \mu m^3$ (half-scan). Mice were injected with 30 μL of 0.5 M solution of Dotarem (Guerbet, France) during repetitive image acquisitions[56].

### Image analysis
All MRI analyses were performed blinded to the experimental data. Analyses of the MCA in MRA were performed using the score: 2: normal appearance, 1: partial occlusion, 0: complete occlusion of the MCA.

All $T_2$*-weighted images presented in this study are minimum intensity projections of 4 consecutive slices (yielding a Z resolution of 600 μm), obtained with ImageJ software 4.11(v. 1.52i). Perfusion index was calculated by measurement of the ratio of ipsilateral and contralateral intensities as described previously[57], using an in-house-created macro, and signal intensity ratios were measured by drawing the region of interest in the liver, spleen, kidney, over the signal intensity of paravertebral muscles as reference, in ImageJ (v1.52i). Lesion sizes on $T_2$-weighted images and signal void on $T_2$*-weighted images were quantified using Slicer software. Signal void quantification on 3D $T_2$*-weighted images, and 3D representation of PHySIOMIC-induced hyposignal were realized using the segmentation and thresholding module on Slicer software (v4.11). Results are presented as volume of particles-induced signal void, or area of signal void (in mm³).

### Two-photon intravital microscopy
To conclude on melamine fluorescent microparticles similarity with PHySIOMIC and observe in vivo *clot* targeting, we injected melamine and melamine-COOH (100 μL) in mice after inducing stroke with thrombin injection, and 100 μL DiOC₆ (1 mg kg⁻¹, Abcam) to reveal platelets and thrombi. Mice were then kept anesthetized during all procedures: the skin of the top of the head was incised and we performed a craniectomy of the right parietal bone. The mice were placed in a stereotaxic frame, and an aqueous medium was deposed between the brain and the ×20 immersive objective. The two-photon excitation wavelength was set at 920 nm, and the resolution at 1.023 pixels per μm (Ultima 2Pplus Microscope, Bruker). Images were acquired using PrairieView Imaging (v.5.7) software.

### In vivo biodegradation of PHySIOMIC and SPIO
The PHySIOMIC and SPIO particles were individually reconcentrated to a concentration of 0.8 mg[Fe] mL⁻¹ using a magnet. They were then intravenously injected to attain a final iron concentration of 4 mg kg⁻¹, with -200 μL of solution administered to each naïve male Swiss mouse (8 weeks, $n = 5$ per group). The ratio polydopamine to iron is of 3.2 so an iron dose of 4 mg kg⁻¹ correspond to a 16,8 mg dose of particles injected per kg. At different time after injection (1 hour, 2, 7 and 31 days), $T_2$-weighted images (RARE sequence, FOV: 4 × 4 mm, TR/TE = 3000 ms/50 ms) were acquired on a BioSpec 7-T TEP-MRI system with a volume coil resonator (Bruker, Germany). At each steps, mice were deeply anesthetized with 5% isoflurane (in a 70/30 gaz mixture N2O/O2) and transcardiacly perfused with saline. Organs of interest (lungs, liver, kidneys, spleen) were harvested for transmission electron microscopy.

### In vitro degradation of PHySIOMIC and SPIO
Four distinct types of relevant buffer solutions were prepared to assess the biodegradation of particles: phosphate-buffered saline (PBS), artificial lysosomal fluid (ALF), citrate buffer (20 mM, pH 4.0) and citrate and hydrogen peroxide (20 mM citrate + $H_2O_2$ 3%)[58]. ALF was prepared with sodium chloride (3.210 g), sodium hydroxide (6.000 g), citric acid (20.800 g), calcium chloride (0.097 g), sodium phosphate heptahydrate (0.179 g), sodium sulfate (0.039 g), magnesium chloride hexahydrate (0.106 g), glycerin (0.059 g), sodium citrate dehydrate (0.077 g), sodium tartrate dehydrate (0.090 g), sodium lactate (0.085 g), and sodium pyruvate (0.086 g) were dissolved in 200 ml of purified water to obtain a 5× stock solution, which was later diluted with purified water and particles during incubation. PHySIOMIC and SPIO were incubated in each buffer at an equivalent iron concentration (0.417 mg mL⁻¹) and at 37 °C. At specific time intervals (0, 3 hours, 24 hours, 3 days, and 7 days), images were captured to document the appearance of the particles in the respective buffers. Additionally, a 1 μL sample was collected for spectral analysis using a UV-Vis spectrophotometer (NanoPhotometer® N60/N50, Implen).

## Hemolysis assay

Erythrocytes were isolated from healthy human whole blood through centrifugation ($2000 \times g$ during 5 minutes) and washed three times with physiological serum. The dispersed erythrocytes were combined with physiological serum at a concentration of 1:10, and PHySIOMIC was introduced to achieve concentrations equivalent to 20, 40, or 80 µg mL$^{-1}$. Negative controls involved the absence of particles, while positive controls consisted of replacing physiological serum with distilled water.

## Clot lysis assay

To evaluate PHySIOMIC impact on clot lysis, we experienced clot lysis time in vitro, using human plasma diluted 1:2 in Hepes buffer [10 mM Hepes, 150 mM NaCl, and 0.4% BSA (pH 7.4)]. Either PHySIOMIC (80 µg[Fe] mL$^{-1}$) or PHySIOMIC with rtPA (1 nM or 5 nM, ACTILYSE®) were incubated with plasma at 37 °C, and changes in plasma opacity were monitored over a period of 12 hours (measuring absorbance at 405 nm) in a microplate reader (FLUOstar Optima, BMG Labtech). Calcium chloride (25 mM) was added to human plasma at the beginning of the experiment to induce clot formation. Clot lysis time was calculated as the duration between the initiation of clot formation and the reduction of absorbance to 50% of its maximal value. All experiments were performed in triplicate.

## Immunohistochemistry

Deeply anesthetized mice were perfused transcardially with saline followed by a fixative solution (4% paraformaldehyde in phosphate buffer) at a physiological rate (8 mL min$^{-1}$) with a peristaltic pump. Brains and liver were post-fixed (24 h, 4 °C) and cryoprotected (20% sucrose, 24 h, 4 °C) before freezing in Tissue-Tek (Miles Scientific). Coronal brain sections (10 µm) were co-incubated with anti-fibrinogen/fibrin (1:1000; sheep antiserum againt fibrinogen prepared at the National institute for agronomic research, France), and platelet-specific marker anti-CD41 (1:1000; BD Biosciences, clone: MWReg30, catalog no. 553847). Primary antibodies were revealed using Fab'2 fragments of donkey anti-rat, goat, sheep IgG linked to FITC, TRITC, or DyLight 629 (1:800, Jackson ImmunoResearch, USA). The specificity of immunostainings was checked by showing the absence of staining when primary antibodies were omitted. The PHySIOMIC could also be revealed via reflection of the polydopamine, using 488 nm laser as emission and setting both, the excitation and the emission filter around the laser wavelength (480 nm to 495 nm) in order to detect reflection of the laser. Images were digitally captured using a Leica DM6000 microscope-coupled coolsnap camera, visualized with MetaVue 5.0 software.

## Histological preparation of brain section

Perls' Prussian Blue Iron Kit stains (Leica Biosystems) were used to detect and identify ferric (Fe3+) iron residue of PHySIOMIC particles in brain sections (10 µm). Images were further acquired with a VS120 Virtual Slide Microscope (Olympus) with VS-AW-L100 software, and processed using QuPath (v0.5.1) and ImageJ (v.1.52i).

## Statistical analysis

All results are presented as mean ± standard deviation (SD). Statistical analyses were performed blinded to the experimental groups, using Graph Pad Prism software (v8.0). We assumed normality of the data distribution with Shapiro–Wilk test. Independent samples $t$ tests or two-sided paired $t$ test were used for comparisons between two groups. Non-parametric data were analyzed by Mann–Whitney, Wilcoxon tests. One-way ANOVA, Kruskal–Wallis and Friedman tests, with Tukey's multiple comparisons test were used for comparing more than two groups. Linear regression was used to model relationship between variables. Differences were considered statistically significant if probability value was inferior to 0.05 ($p < 0.05$).

## Reporting summary

Further information on research design is available in the Nature Portfolio Reporting Summary linked to this article.

## Data availability

The data supporting the findings from this study are available within the manuscript and its supplementary information. Proteomics data have been deposited in iProX under accession code IPX0008212001 https://www.iprox.cn//page/. All magnetic resonance imaging and confocal files are not provided as the size is too important, but any raw data will be available from the corresponding author upon reasonable request. Source data are provided in this paper.

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

## Acknowledgements

This work was supported by the Agence Nationale de la Recherche "PHySIOMIC" ANR-20-CE19-0031 (to T.B. and A.P.); "FLAMRING" ANR-20- CE19-0032-01 (to S.M.d.L.); the Region Normandie under the convention RIN Doctorant project Modirem (to C.J.); and "Fondation Bettencourt Schueller" for a CCA-Inserm-Bettencourt position (to M.G.). We kindly thank Marion Berthelot from the PhIND unit for providing access to the plasma samples and Dr. Nicolas Elie from U.S Emerode, Université de Caen Normandie, Federative Structure 4207 'Normandie Oncologie',

PLATON Services Unit, VIRTUAL'HIS for granting access to the slide scanner and offering his advice. The slide scanner VS120 was purchased through the European project "ONCOTHERA" is funded by the European Union within the framework of the Operational Program ERDF/ESF 2014-2020.

## Author contributions

The authors C.J., A.P., J.F., M.R., F.L., S.M.d.L., A.M., F.P., M.N., B.B., D.G, and T.B. have contributed to experimentations. Supervision of the project was ensured by D.V. and T.B. Conceptualization of the experiments was done by C.J., A.P., P.M., M.N., S.M.d.L., A.M-F., B.B., M.G., D.V and T.B. The authors C.J. and T.B. contributed to the writing of the manuscript.

## Competing interests

The authors declare the following competing interests: S.M.d.L., C.J., D.V., T.B., and M.G. have filed a patent application (WO2023007002A1) for the use of PHySIOMIC as a contrast agent to reveal microthrombi in ischemic stroke. The other authors declare that they have no competing interests.
