## [Peer Review File · Nature Communications]

REVIEWER COMMENTS

Reviewer #1 (Remarks to the Author):

This manuscript evaluates a unique MRI based contrast agent (PHySIOMIC) for detection of microthrombi in the brain during experimental stroke.

I have a number of questions/concerns/suggestions about the experiments performed.

Page 20, paragraph 2: etiology is misspelled.

Was the parent vessel still occluded at the time hypointense signal was seen after PHySIOMIC injection? 3D TOF MRI should be able to confirm such? Could PHySIOMIC be simply measuring poor cerebral blood flow from a proximal occlusion rather than true microthrombi?

Please provide more details in the pathologic analysis of how the PHySIOMIC is aggregating with microthrombi? Figure 2g. Magnified images of the clusters of iron oxide and their association with microthrombi (stained by specific immunostains to confirm such).

Please show some electron microscopic evidence of PHySIOMIC/iron oxide particles in proximity to microthrombi from brain samples at the location of the infarct where microthrombi seen. EM was performed for spleen and liver deposition of PHySIOMIC so could also be done for brain samples.

Why does the AICI3 model (Figure 3) not produce microthrombi?

Figure 6 is assessing Melamine and Melamine-Carboxylic not PHySIOMIC?

Supplemental Figure 4 raises major concerns about the MR contrast agent slowing the effects of clot lysis with TPA with longer clot lysis times. Please discuss in Discussion section.

Supplemental Figure 5: Authors should speculate whether the disappearance of PHySIOMIC with time is a product of the half life/metabolism properties of the agent or that microthrombi clear over time?

Supplemental Figure 6d: The Time of Flight image provided offers no compelling evidence of recanalization. The wrong side of the brain has an arrow?

Please orient the reader to which side of the brain we are seeing for each Figure of the manuscript. Is Left on left and right on right consistently for the various imaging figures provided throughout the manuscript?

Please consider showing a more representative example of persistent occlusion versus recanalization at 6 hours.

Reviewer #2 (Remarks to the Author):

This manuscript describes a new MRI contrast agent, PHySIOMIC, which is intended to allow MR imaging of thrombi and microthrombi in a mouse model of thromboembolic ischemic stroke. Currently there is no accurate way of imaging thrombi and microthrombi in the brain. Being able to do so, would have tremendous value in both detecting such lesions during stroke, and also monitoring the response of individual sites following standard of care treatments. The significance of this work is therefore very high. Detection of both thrombi and microthrombi and monitoring of thrombolytic treatment was demonstrated in this mouse model. The authors present very well-designed experiments to assess the targeting and MR imaging characteristics, biologic mechanism of action, degradation, and clearance pathways, as well as the ability of the agent to monitor the efficacy of response in near real time which is imperative in stroke patients. The approach was very thorough, and data provided strongly supported the conclusions. Statistical support is appropriate. Supplementary materials all appear to be important for supporting conclusions. There are only a few questions or comments that would improve the paper which follow.

1. It is stated in the methods section on p26 that either "PHySIOMIC or SPIO particles with concentrations of 4 mg [FE]/kg were administered to naïve male Swiss mice. Please clarify this dosing

nomenclature and provide a total mass dose and volume of each agent administered per mouse. Please provide clear dosing masses and volumes when agents are injected. Although assessing safety and tolerance of the agent was not an aim of this paper, it will be necessary to clearly understand the dosing especially since large volumes of particles greater than 500 nm in diameter have proven problematic in early human studies [Ivancev, et al. (1989) Effect of Intravenously Injected Iodinated Lipid Emulsions on the Liver, *Acta Radiologica*, 30:3, 291-298, DOI: 10.3109/02841858909174683] where it was found that the sinusoids in the liver became bloated ultimately blocking the flow of bile causing significant lower back pain and ultimately precluding the use of these particulates in humans.

2. Page 6: line 4, change "...phantoms using a 7T..." to "phantoms using 7T..." and line 28, change "...major risk associated..." to "...major risks associated..."

3. A burning question is whether or not these 750 nm particles would be rapidly sequestered by the RE cells in the liver and if MRI of the liver would show significant uptake. This question was beautifully addressed on p 16. It may be prudent to add a sentence or two on the potential impact of significant first pass liver clearance on the amount of agent ultimately reaching the micro-thrombi in the brain and whether or not alternative injection sites might allow more efficient targeting.

4. Page 22, Transmission Electron Microscopy section: line 1: change dry to dried. Line 7: change ultrathion to ultrathin.

5. Page 24 Exclusion criteria section: line 1, change MIE to Mice

6. Page 26: Please justify why only male mice were used in this study. If it is because of their large size in this strain, then say this.

Reviewer #3 (Remarks to the Author):

The paper describes the development of magnetic material based on iron oxide and polydopamine that enables the non-invasive in vivo diagnosis and monitoring of microthrombi using MRI and during acute ischemic strokes. The ability of the material was assessed on a mouse model by identifying differences in MRI response and by confirming them by histological analysis. The results were associated with a study of the protein corona, which indicated the key role of fibrinogen in the corona in the targeting to microthrombi. Finally, the authors studied the fate of the injected contrast agent, which is degraded in the liver and spleen. The relevance of the work is that the described methodology allows the identification of microvascular thrombosis through non-invasive methods, which is a challenging and important task because the microthrombi are associated with poorer clinical outcomes.

As for the specific part associated with my expertise, the corona appears to have a fundamental role in targeting. Indeed, the authors characterize the corona because no active targeting by ligands is present on the material. Fibrinogen is used to explain the targeting.

- 1) The authors could add a brief explanation as to why fibrinogen in the corona can target clots.
- 2) A brief discussion on the role of other major proteins in the corona can also be added, to elucidate possible interactions with the immune system. Please focus on the PHySIOMIC particles rather than melamine particles.
- 3) A quantitative analysis of fibrinogen in PHySIOMIC and melamine particles could also be included, to support the results with the commercial fluorescent particles.
- 4) In the methods section, please add a sentence on the type of processing used for proteomic analysis, with a reference to the protocol.
- 5) Figure 7 in supplementary material is difficult to read, please improve size and font. In addition, please provide the corona composition as an Excel file with matched protein groups and associated information for identification.
- 6) Deposit in a public repository of raw LC/MS data and protein identifications could be done.

Reviewer #4 (Remarks to the Author):

In this study, the authors have developed a contrast agent of MRI called PHySIOMIC, synthesized through a self-assembly process of SPIO within a polydopamine matrix. This approach has enabled us to combine the advantages of clinically approved SPIO, which possesses good T2* relaxivity in MRI, with those of micro-sized particles. These micro-sized particles enhance T2* contrast and enable rapid absorption for fast molecular imaging. Additionally, the polydopamine matrix conferred thrombosis targeting properties that enabled molecular MRI of microthrombi in a thrombo-embolic ischemic stroke murine model. Overall, this design is very intriguing. However, there are a few concerns that need to be clarified before consideration for publication.

1. The authors found that PHySIOMIC without any ligands targeted microthrombi through the formation of the protein corona. However, the effect of the particle size and zeta potential of PHySIOMIC on the microthrombi targeting was not well revealed. In addition, the composition and the underlying mechanism of the protein corona to facilitate PHySIOMIC's microthrombi targeting were not well discovered.

2. Figure 2b demonstrated that the imaging effect of PHySIOMIC was significantly superior to that of SPIO. It was proven that the protein corona of PHySIOMIC could enhance its microthrombi-targeting. However, there is no additional analysis of SPIO to illustrate the unique composition of the protein corona of PHySIOMIC. Please provide further evidence.

3. The formation of the protein corona on nanoparticles could accelerate the removal of particles from the bloodstream, thereby reducing the amount of PHySIOMIC reaching the microvasculature. It was also evident that the half-life of PHySIOMIC was much shorter than that of SPIO. Why did PHySIOMIC have a superior targeting to microthrombi? Please discuss the results.

4. Figure 7a illustrated that both PHySIOMIC and SPIO particles induced a significant decrease in signal intensity in the liver and spleen, indicating rapid clearance of the particles from the body. Additionally, in Fig. 7b, it is noteworthy that the signal intensity after injection is considerably lower than before, and it gradually increases with time. Can you explain the underlying cause of this phenomenon?

5. The administration of thrombolysis with rtPA results in increased permeability of the blood-brain barrier. However, using PHySIOMIC at this stage is associated with an elevated risk of the contrast agent crossing the blood-brain barrier, which can potentially cause severe brain injury. Please discuss the advantages and disadvantages of PHySIOMIC as an MRI contrast agent.

6. It is advisable to carefully proofread the manuscript for any errors in grammar and typos.

Dear Reviewers

Title: MRI-Based Microthrombi Detection in Stroke with Polydopamine Iron Oxide

Manuscript - NCOMMS-23-56845-T

We would like to thank the reviewers for their recommendations.

In response to the comments, we have enriched the manuscript following the suggestions of the 4 reviewers.

In total, we have modified 8 figures (Figures 2, 3, 4, 5, 6, S7, S8, and S9), added 2 supplementary figures (Figures S5 and S6) and 2 other supplementary materials; a video file (supplementary material 1) and the detailed analysis of raw data (supplementary material 2).

We enriched the introduction, method, results, and discussion sections to answer the different comments. We also shortened the abstract to respect the 150-word limit and shortened the subheadings as requested.

All modifications are listed below and new text is highlighted in blue. Additionally, a version of the manuscript with all modifications identified via the tracking mode is provided; it shows in red the modifications. A clean final version of the revised manuscript is also provided.

We hope the modifications will meet the requirements of the editor and the different reviewers.

With best regards,

Corresponding Author

Reviewer #1 (Remarks to the Author):

This manuscript evaluates a unique MRI-based contrast agent (PHySIOMIC) for detection of microthrombi in the brain during experimental stroke.

I have a number of questions/concerns/suggestions about the experiments performed.

Page 20, paragraph 2: etiology is misspelled.

We have corrected this error, thank you for spotting it.

Was the parent vessel still occluded at the time hypointense signal was seen after PHySIOMIC injection? 3D TOF MRI should be able to confirm such? Could PHySIOMIC being simply measuring poor cerebral blood flow from a proximal occlusion rather than true microthrombi?

In both ischemic stroke models presented in Figure 3, we confirm that the parent vessel is still occluded at the time of the injection as confirmed by the TOF as well as perfusion MRI (Figure 3d). We also confirm that this blood flow deficit induces tissue injury as we detect the formation of a brain infarct via diffusion-weighted MRI. In both models, the consequences in terms of blood flow reduction are similar. Yet, the results are radically different between the two models; in the thrombin model we observe a strong signal corresponding to PHySIOMIC accumulation and we can observe microthrombi in histology sections, while in the AlCl₃ model, we have no signal and no microthrombi in histology (figure 3d). If the PHySIOMIC were measuring poor cerebral blood flow, a similar signal would have been obtained in both models. Moreover, in every observation of histology sections (TEM, immuno-histology, or PERLS staining) or two-photon microscopy, we found the PHySIOMIC exclusively where thrombi were found. We now provide more data in terms of histology and added TEM observations in order to clarify this point (Supplementary Figures 5 and 6, more details are provided in the following questions).

However, it is not to be excluded that the slower blood flow downstream of the proximal clot contributes to the effective binding of our particles to the microthrombi. The decrease in blood flow might enhance the contact between the particles and the microthrombi, thereby facilitating binding.

Please provide more details in the pathologic analysis of how the PHySIOMIC is aggregating with microthrombi? Figure 2g. Magnified images of the clusters of iron oxide and their association with microthrombi (stained by specific immunostains to confirm such).

Regarding the analysis of how the PHySIOMIC is binding to the microthrombi, we hypothesize that the fibrinogen content from the protein corona confers affinity to activated platelets.

We now have a whole set data presented in Figure 6 further supporting this hypothesis. Accordingly, we added several sentences to the revised version of the manuscript. We discussed this hypothesis in the discussion section, p 19-20:

“we notably investigated the composition of the PC of PHySIOMIC particles and pursued the hypothesis that its fibrinogen composition could contribute to the targeting process to microthrombi. However, it remains plausible that other PC components might contribute to the thrombosis targeting phenomenon. In such conditions, it should be noted that the presence of a higher concentration of fibrinogen in the protein corona on the PHySIOMIC particles could also have prothrombotic consequences. At this stage, we did not completely elucidate to which specific thrombosis component or protein the protein corona confer adhesion, but one possible target could be the activated form of platelet integrin α IIb β 3 which is known to specifically recognize fibrinogen³⁸. This hypothesis is in accordance with the transmission electronic microscopy observations as the PHySIOMIC were found attached to activated platelets.”

Regarding Figure 2g, we removed the histology observations to put forward the TEM observations (explained in the next question). We also added a supplementary figure exclusively dedicated to histology and immune-staining of microthrombi. We now provide images from histology sections of a brain harvested after the intravenous injection of PHySIOMIC, just after the MRI procedure, where the microthrombi were revealed via CD41 and where the PHySIOMIC could be detected by light reflection on the surrounding of the microthrombi. The Perls' staining also supports the fact that the iron oxide clusters are localized around the microthrombi.

We added an explanation of this additional observation in the method section, p 27:

“The specificity of immunostainings was checked by showing the absence of staining when primary antibodies were omitted. The PHySIOMIC could also be revealed via reflection of the polydopamine, using 488nm laser as emission and setting both, the excitation and the emission filter around the laser wavelength (480nm to 495nm) in order to detect reflection of the laser.”

We completed the results p 7:

“We can conclude that, in this model, PHySIOMIC particles constitute a diagnosis tool to reveal microthrombi and to predict the size of the ischemic lesions. Microthrombi could be identified via platelet marker CD41 immunolabelling in the ischemic area on histological analysis of brains harvested just after the MRI acquisition, 1 hour after the stroke induction. The PHySIOMIC particles could be detected via light reflection from polydopamine positioned around thrombi and the microthrombi, on the luminal side of blood vessels (Supplementary Fig. 5a,b). Perls staining confirmed this specific localization of the particles, stained blue due to their iron oxide composition (Supplementary Fig. 5c).”

See below the added Figure S5:

Please show some electron microscopic evidence of PHySIOMIC/iron oxide particles in proximity to microthrombi from brain samples at the location of the infarct where microthrombi seen. TEM was performed for spleen and liver deposition of PHySIOMIC so could also be done for brain samples.

We would like to thank the reviewer for this interesting suggestion. The requested experiments were performed with the help of the electron microscopy imaging facility MicroEleCS from the INSERM UMR-S1255 Unit in Strasbourg. The images obtained following the processing of the samples are presented in the new Supplementary Figure 6

We added the description of the experiment in the method section p 23:

“Microthrombi in brain section were observed at the electron microscopy imaging facility MicroEleCS (INSERM UMR-S1255 Unit, Strasbourg, France). Brain tissue was fixed with 2.5% glutaraldehyde in 0.1 M sodium cacodylate buffer containing 2% sucrose, previously warmed to 37°C (305 mOsm, pH 7.3) for 1 hour. The samples were then rinsed and postfixed with 1% osmium tetroxide in 0.1 M sodium cacodylate buffer for 1 hour at 4°C. After additional washing in the 0.1 M sodium cacodylate buffer, the samples were dehydrated in successively increasing ethanol concentrations before embedding in epoxy. The resin was polymerized for 2 days at 50°C. Ultrathin sections (100 nm) were stained with lead citrate and uranyl acetate, and examined under a Jeol 2100-plus (120 kV).”

In the result section p 8:

“To further investigate the localization of the PHySIOMIC within the microthrombi, we performed transmission electronic microscopy (TEM) of the brain sections and we observed the iron oxide clusters close to degranulated platelets (Fig. 2g and Supplementary Fig 6). In all observations of the histological sections, the PHySIOMIC particles were identified inside or at the surface of microthrombi, without noticeable accumulation in other parts of the brain.”

See below the added Figure S6:

Why does the AlCl₃ model (Figure 3) not produce microthrombi?

AlCl₃ is deposited locally over the MCA bifurcation and enables thrombosis formation restricted to the vessel exposed to AlCl₃. In contrast, in the thrombin model, recombinant murine thrombin is infused intra-arterially and reach not only the main artery where the pipet tip is located, but also the whole downstream microcirculation, thereby leading to microthrombi formation. We completed the result part with more description of the model and an explanation of this specific difference, p10:

*“We further compared the results obtained after PHySIOMIC injection in this first ischemic stroke model induced via thrombin injection in the MCA (referred to as ‘thrombin model’), to a second ischemic stroke model in which thrombosis was induced *in situ* via local deposition on the MCA of a filter paper soaked with aluminum chloride (Fig 3). In this second model (referred to as the ‘AlCl₃ model’), a unique clot is formed precisely at the area in contact with AlCl₃ and no embolization in the microcirculation occurs.”*

Figure 6 is assessing Melamine and Melamine-Carboxylic not PHySIOMIC?

Indeed, in Figure 6, we focused mainly on fluorescent particles of Melamine and Melamine-Carboxylic except for the results from the mass spectrometry for the analysis of the plasma protein corona where we also performed the experiment on the PHySIOMIC. The reason we could not perform the same experiments on the PHySIOMIC is that we were not able to label them with fluorescent dye as the polydopamine modifies optic properties.

We explain this in the discussion section, p 21:

*“Due to the light-absorbing properties of polydopamine, it was not possible to obtain fluorescently labeled polydopamine particles without surface modification for *in vitro* experiments. Therefore, to delve deeper into the thrombosis targeting mechanism identified with PHySIOMIC particles, we opted to use commercially available fluorescent particles composed of melamine resin. This choice was motivated by the presence of amine groups on the surface of melamine resin, which is akin to polydopamine. Furthermore, it has been reported that modifying the external layer of nanomedicines with carboxylate groups significantly influences the profile of adsorbed proteins compared to bare particles with amine groups³⁶. Consequently, we also used melamine functionalized with carboxylic termination (melamine-COOH). Thanks to the strong fluorescent labeling of melamine and melamine-COOH, we were able to track them *in vitro* through a microfluidic study and *in vivo* with two-photon microscopy.”*

Supplemental Figure 4 raises major concerns about the MR contrast agent slowing the effects of clot lysis with tPA with longer clot lysis times. Please discuss in Discussion section.

The supplementary figure 4 indicates that the presence of the MR contrast agent might slightly slow the clot lysis with tPA. Although this difference is not significant for both the tPA concentrations tested, we agree that this is an important risk to consider for future development. In addition, the thrombosis targeting mechanism that we elucidated in this study involves the presence of fibrinogen in higher concentration in the protein corona of the particles adhering to thrombosis which in itself may carry prothrombotic risk. We propose to raise this aspect in the discussion p19-20:

“In such conditions, it should be noted that the presence of a higher concentration of fibrinogen in the protein corona on the PHySIOMIC particles could also have prothrombotic consequences. At this stage, we did not completely elucidate to which specific thrombosis component or protein the protein corona confer adhesion, but one possible target could be the activated form of platelet integrin α IIb β 3 which is known to specifically recognize fibrinogen³⁸. This hypothesis is in accordance with the transmission electronic microscopy observations, where the PHySIOMIC were found attached to activated platelets. So, in addition to bringing some fibrinogen to the edge of the clot and thus possibly enhancing fibrin

polymerization, the PHySIOMIC covered with fibrinogen could also increase platelet aggregation. Besides, although the clot lysis assay did not show a significant difference in the tPA-induced clot lysis time in the presence of PHySIOMIC, the mean value of lysis time was numerically higher. Overall, the putative risk of prothrombotic effect should be monitored for further developments of the PHySIOMIC particles.”

Supplemental Figure 5: Authors should speculate whether the disappearance of PHySIOMIC with time is a product of the half life/metabolism properties of the agent or that microthrombi clear over time?

The disappearance of the signal coincides with recanalization time and for this reason, we attribute the loss of signal to microthrombi clearance rather than proper half-life/metabolism of the PHySIOMIC. Indeed, we observe disappearance of PHySIOMIC on MRI over a few hours, whereas usually the metabolism of iron oxide nanoparticles requires a few weeks to reverse the MRI signal (Martinez de Lizarondo et al., Science Advances 2022). We added a sentence to precise this hypothesis, p12:

While the signal void was consistent with previous observations when measured at early time points after stroke onset (30 minutes and 2 hours post-occlusion), it almost completely disappears at 6 hours post-occlusion (Supplementary Fig. 8c). These events coincide with cerebral reperfusion, as indicated by the increase in the angiographic score (Supplementary Fig. 8d,e) **which supports that this slow progressive signal disappearance is caused by the clearance of the microthrombi rather than *in situ* metabolism of the PHySIOMIC.”**

Supplemental Figure 6d: The Time of Flight image provided offers no compelling evidence of recanalization. The wrong side of the brain has an arrow? Please orient the reader to which side of the brain we are seeing for each Figure of the manuscript. Is Left on left and right on right consistently for the various imaging figures provided throughout the manuscript?

Time-of-flight images are presented along the axial axis, viewed from above. In this representation, the MCA was observed on the right of the images (where was the arrow). To ensure better comprehension, we changed the visualization of the TOF images to place the MCA on the left in all images. The TOF images are seen now from below. We also added left and right indications on the images (noted L & R).

Please consider showing a more representative example of persistent occlusion versus recanalization at 6 hours.

The images used as examples to demonstrate recanalization are not the most representative; they have been modified to better match the average of the data presented in the recanalization graph. These modifications also led us to change the orientation of the images in Figure 3 (perfusion imaging and angiography) to correspond.

Reviewer #2 (Remarks to the Author):

This manuscript describes a new MRI contrast agent, PHySIOMIC, which is intended to allow MR imaging of thrombi and microthrombi in a mouse model of thromboembolic ischemic stroke. Currently there is no accurate way of imaging thrombi and microthrombi in the brain. Being able to do so, would have tremendous value in both detecting such lesions during stroke, and also monitoring the response of individual sites following standard of care treatments. The significance of this work is therefore

very high. Detection of both thrombi and microthrombi and monitoring of thrombolytic treatment was demonstrated in this mouse model. The authors present very well-designed experiments to assess the targeting and MR imaging characteristics, biologic mechanism of action, degradation, and clearance pathways, as well as the ability of the agent to monitor the efficacy of response in near real time which is imperative in stroke patients. The approach was very thorough, and data provided strongly supported the conclusions. Statistical support is appropriate. Supplementary materials all appear to be important for supporting conclusions. There are only a few questions or comments that would improve the paper which follow.

1. It is stated in the methods section on p26 that either “PHySIOMIC or SPIO particles with concentrations of 4 mg [FE]/kg were administered to naïve male Swiss mice. Please clarify this dosing nomenclature and provide a total mass dose and volume of each agent administered per mouse. Please provide clear dosing masses and volumes when agents are injected. Although assessing safety and tolerance of the agent was not an aim of this paper, it will be necessary to clearly understand the dosing especially since large volumes of particles greater than 500 nm in diameter have proven problematic in early human studies [Ivancev, et al. (1989) Effect of Intravenously Injected Iodinated Lipid Emulsions on the Liver, *Acta Radiologica*, 30:3, 291-298, DOI: 10.3109/02841858909174683] where it was found that the sinusoids in the liver became bloated ultimately blocking the flow of bile causing significant lower back pain and ultimately precluding the use of these particulates in humans.

The sentence in the Materials and Methods indicating the dose administered to mice was changed for a better understanding of the injected dose, p29:

The PHySIOMIC and SPIO particles were individually reconcentrated to a concentration of 0.8 mg[Fe].mL⁻¹ using a magnet. They were then intravenously injected to attain a final iron concentration of 4 mg.kg⁻¹, with approximately 200µL of solution administered to each naïve male Swiss mouse (8 weeks, n = 5 per group). The ratio polydopamine to iron is of 3.2 so an iron dose of 4 mg.kg⁻¹ correspond to a 16,8 mg dose of particles injected per kg.

Concerning the risks associated with particles larger than 500 nm, we thank the reviewer for pointing us to this study and the specific risks of large particles. We agree that it will be crucial to conduct further stability/toxicity studies before advancing these particles to clinical trials to ensure their safety. Knowing this specific risk of liver sinusoid embolization will be useful at this point and we will certainly add it to the list of risk to be considered.

*From our side, we performed a more comprehensive toxicity study regarding the risk in the liver with similar particles in a study we published recently (Sara Martinez de Lizarrondo et al., Tracking the immune response by MRI using biodegradable and ultrasensitive microprobes. *Sci. Adv.* 8, eabm3596(2022). DOI:10.1126/sciadv.abm3596). We verified the non-toxicity of other submicrometric iron oxide-based particles, with a 6-month follow-up on animals. We observed hepatic biodegradation of the particles, assessed the condition of the liver, kidneys, lungs, and spleen via histology, and verified the absence of ASAT and ALAT or pro-inflammatory cytokines generated. With this data, we are confident that iron oxide polydopamine particles with a diameter between 500nm and 1 µm do not induce toxic side effects if injected at reasonable dose. It should however be acknowledged that preclinical studies cannot reveal every side effects and that consequences of intravenous injection of such particles in human remain largely unknown.*

2. Page 6: line 4, change “...phantoms using a 7T...” to “phantoms using 7T...” and line 28, change “...major risk associated...” to “...major risks associated...”

Thank you for spotting these mistakes, we corrected them.

3. A burning question is whether or not these 750 nm particles would be rapidly sequestered by the RE cells in the liver and if MRI of the liver would show significant uptake. This question was beautifully addressed on p 16. It may be prudent to add a sentence or two on the potential impact of significant first pass liver clearance on the amount of agent ultimately reaching the micro-thrombi in the brain and whether or not alternative injection sites might allow more efficient targeting.

We added a specific paragraph in the revised version of the discussion to discuss the potential negative impact of the short half-life. We agree that the number of particles reaching the target is most likely very low (probably around 0.5%). However, it should also be noted that if the particles were decorated with PEG chains to decrease the RE sequestration, for example, the targeting would not be effective anymore because of a perturbed protein corona. Regarding the injection route, it is indeed possible that intra-arterial injection would increase the targeting ratio as the particles would pass through the brain before being filtered by the liver.

We now discuss this aspect in the discussion:

It is also interesting to note that the PHySIOMIC provided strong imaging signal detection although they exhibited an extremely short half-life (less than a minute) compared to the SPIO which typically circulates much longer (over 10 minutes) but was not able to reveal the microthrombi. This might appear surprising as the field of nanomedicine usually supports long circulation half-life as a positive feature for molecular imaging applications as longer exposure time should logically result in increased targeting³⁵. However, this study provides evidence that this reasoning is not always true and that, at least in some contexts, modifying the surface to improve the targeting properties can be more valuable than to increase the circulation time.

4. Page 22, Transmission Electron Microscopy section: line 1: change dry to dried. Line 7: change ultrathion to ultrathin.

Thank you for spotting these mistakes, we corrected them.

5. Page 24 Exclusion criteria section: line 1, change MIE to Mice

Thank you for spotting this mistake, we corrected it.

6. Page 26: Please justify why only male mice were used in this study. If it is because of their large size in this strain, then say this.

We generally use exclusively male mice because we encounter more reproducibility as we avoid the potential impact of the estrous cycle. This is a general habit of preclinical research and animal experiments that we are aware is not ideal, as we might miss an effect related to sex. For this reason, in this study, we have now included a study on female mice to study the potential effect of sex (cf described in the answer to editor comments). In summary, we do not observe any significant differences between male and female mice for microthrombi imaging using PHySIOMIC.

Reviewer #3 (Remarks to the Author):

The paper describes the development of magnetic material based on iron oxide and polydopamine that enables the non-invasive in vivo diagnosis and monitoring of microthrombi using MRI and during acute ischemic strokes. The ability of the material was assessed on a mouse model by identifying differences in MRI response and by confirming them by histological analysis. The results were associated with a study of the protein corona, which indicated the key role of fibrinogen in the corona in the targeting to microthrombi. Finally, the authors studied the fate of the injected contrast

agent, which is degraded in the liver and spleen. The relevance of the work is that the described methodology allows the identification of microvascular thrombosis through non-invasive methods, which is a challenging and important task because the microthrombi are associated with poorer clinical outcomes.

As for the specific part associated with my expertise, the corona appears to have a fundamental role in targeting. Indeed, the authors characterize the corona because no active targeting by ligands is present on the material. Fibrinogen is used to explain the targeting.

1) The authors could add a brief explanation as to why fibrinogen in the corona can target clots.

We added a hypothesis on this explanation in the discussion section p 19-20:

“At this stage, we did not completely elucidate to which specific thrombosis component or protein the protein corona confer adhesion, but one possible target could be the activated form of platelet integrin $\alpha\text{IIb}\beta\text{3}$ which is known to specifically recognize fibrinogen³⁸. This hypothesis is in accordance with the transmission electronic microscopy observations, where the PHySIOMIC were found attached to activated platelets.”

2) A brief discussion on the role of other major proteins in the corona can also be added, to elucidate possible interactions with the immune system. Please focus on the PHySIOMIC particles rather than melamine particles.

To enrich the discussion, we have incorporated a paragraph in the Discussion section addressing the proteins found in the protein corona and their potential influence on PHySIOMIC particles' biodistribution, blood half-life, and immune response, p19:

“It is noteworthy that the protein corona plays a significant role in the biodistribution, blood circulation time, and immune response to particles. Thus, the presence of opsonins within this protein corona facilitates rapid recognition by the complement system and the clearance of particles from circulation by macrophages.”

3) A quantitative analysis of fibrinogen in PHySIOMIC and melamine particles could also be included, to support the results with the commercial fluorescent particles.

To address this question, we conducted an additional experiment, as depicted in the new Supplementary Figure 7, to measure fibrinogen adsorption on melamine and PHySIOMIC particles using a Nanodrop spectrophotometer (Implen). A fibrinogen solution (1 mg/mL) was exposed to the different particles, and after centrifugation, the particles were separated from the solution, allowing us to measure the fibrinogen content in the supernatant. The results support our hypothesis that fibrinogen adsorption is more pronounced in PHySIOMIC and melamine particles targeting microthrombi. In contrast, fibrinogen absorption was found to be negligible on melamine-COOH particles. Unfortunately, it was not possible to measure PHySIOMIC-BSA's fibrinogen adsorption using this technique due to interference between BSA and fibrinogen spectra on the nano spectrophotometer. We have included this experiment in the Results section, along with the technique in the Methods section.

Results section, p 16: “Remarkably, we noticed an augmentation in the number of particles adhering to thrombi when the particles were suspended in fibrinogen-rich plasma (Fig. 6f). Hence, the fibrinogen integrated into the PCs of PHySIOMIC while in circulation plays a pivotal role in their passive targeting to thrombi and microthrombi. To measure the adsorption of fibrinogen onto the surface of particles, we exposed the particles to a fibrinogen solution (1 mg.mL⁻¹) during 30 minutes at 37°C. The particles were then centrifuged, and the fibrinogen remaining in the supernatant was measured using a nano spectrophotometer. Fibrinogen adsorption is greater on melamine and PHySIOMIC particles, with mean

values of 50 μg and 75 μg , respectively, per 250 μl of particles (Figure 6g).”

Methods section, p26:

“Fibrinogen adsorption quantification

Melamine (diluted at 1:5), Melamine-COOH (diluted at 1:50) and PHySIOMIC particles were incubated with a fibrinogen solution (1 mg.mL⁻¹) at a 1:1 ratio during 30 minutes at 37°C. The particles were then centrifuged (3 minutes, 10000 rpm) and the supernatant was measured with a nano spectrophotometer (Nanodrop, Implen) with a molar extinction coefficient of 1,51. The amount of fibrinogen adsorbed onto the surface of the particles is measured by subtracting the fibrinogen value measured in the supernatant from the control fibrinogen value (incubation of fibrinogen and mannitol buffer of the particles).”

See below the modified Figure 6:

4) In the methods section, please add a sentence on the type of processing used for proteomic analysis, with a reference to the protocol.

The proteomic protocol conducted by the Proteogen platform has been added to the "Methods" section, p 24-25:

“Five µg of each protein extract were prepared using a modified Gel-aided Sample Preparation protocol⁵¹. Samples were digested with trypsin/Lys-C overnight at 37°C. For nano-LC fragmentation, protein or peptide samples were first desalted and concentrated onto a µC18 Omix (Agilent) before analysis. The chromatography step was performed on a NanoElute (Bruker Daltonics) ultra-high-pressure nano flow chromatography system. Approximately 200ng of each peptide sample were concentrated onto a C18 pepmap 100 (5mm x 300µm i.d.) precolumn (Thermo Scientific) and separated at 50°C onto a reversed phase Reprosil column (25cm x 75µm i.d.) packed with 1.6µm C18 coated porous silica beads (Ionopticks). Mobile phases consisted of 0.1% formic acid, 99.9% water (v/v) (A) and 0.1% formic acid in 99.9% ACN (v/v) (B). The nanoflow rate was set at 300 nl/min, and the gradient profile was as follows: from 2 to 15% B within 15 min, followed by an increase to 25% B within 10 min and to 37% B within 12 min and further to 9% within 7 min and reequilibration. Mass Spectrometry (MS) experiments were carried out on an TIMS-TOF pro mass spectrometer (Bruker Daltonics) with a modified nano electrospray ion source (CaptiveSpray, Bruker Daltonics). A 1400 spray voltage with a capillary temperature of 180°C was typically employed for ionizing. MS spectra were acquired in the positive mode in the mass range from 100 to 1700 m/z and 0.75 to 1.30 1/k0 window. In the experiments described here, the mass spectrometer was operated in PASEF DIA mode with exclusion of single charged peptides. The DIA acquisition scheme consisted of 24 variable windows ranging from 300 to 1000 m/z.”

5) Figure 7 in supplementary material is difficult to read, please improve size and font. In addition, please provide the corona composition as an Excel file with matched protein groups and associated information for identification.

Thank you for your feedback. The layout of the graphs has been adjusted, and their sizes have been increased to enhance clarity in reading the figure. The composition of the protein corona is provided in the Excel file of the Raw data. In this file, we have listed the proteins, their affiliations with a biological cluster, as well as the measurements by mass spectrometry.

6) Deposit in a public repository of raw LC/MS data and protein identifications could be done.

According to your recommendations, the mass spectrometry data has been added to a repository on the iProX website, under the number: IPX0008212001.

Reviewer #4 (Remarks to the Author):

In this study, the authors have developed a contrast agent of MRI called PHySIOMIC, synthesized through a self-assembly process of SPIO within a polydopamine matrix. This approach has enabled us to combine the advantages of clinically approved SPIO, which possesses good T2* relaxivity in MRI, with those of micro-sized particles. These micro-sized particles enhance T2* contrast and enable rapid absorption for fast molecular imaging. Additionally, the polydopamine matrix conferred thrombosis targeting properties that enabled molecular MRI of microthrombi in a thrombo-embolic ischemic stroke murine model. Overall, this design is very intriguing. However, there are a few concerns that need to be clarified before consideration for publication.

1. The authors found that PHySIOMIC without any ligands targeted microthrombi through the formation of the protein corona. However, the effect of the particle size and zeta potential of PHySIOMIC on the microthrombi targeting was not well revealed. In

addition, the composition and the underlying mechanism of the protein corona to facilitate PHySIOMIC's microthrombi targeting were not well discovered.

While the mechanism involved in targeting microthrombi through protein corona formation has yet to be investigated, we demonstrate the importance of fibrinogen in this mechanism using microfluidic experiments. We proposed in a paragraph added to the Discussion Section that PHySIOMIC, with fibrinogen adsorbed on its surface, may bind to platelet integrin $\alpha\text{IIb}\beta\text{3}$ in microthrombi:

“Thus, we notably investigated the composition of the PC of PHySIOMIC particles and pursued the hypothesis that its fibrinogen composition could contribute to the targeting process to microthrombi. However, it remains plausible that other PC components might contribute to the thrombosis targeting phenomenon. In such condition, it should be noted that the presence of a higher concentration of fibrinogen in the protein corona on the PHySIOMIC particles could also have prothrombotic consequences. At this stage we did not completely elucidate to which specific thrombosis component or protein the protein corona confer adhesion, but one possible target could be the activated form of platelet integrin $\alpha\text{IIb}\beta\text{3}$ which is known to specifically recognize fibrinogen³⁸. This hypothesis is in accordance with the transmission electronic microscopy observations as the PHySIOMIC were found attached to activated platelets.”

Regarding the particle size and zeta potential, we agree that they are important parameters with an impact on targeting via the circulation time and via the influence on the protein corona. We provide a table to list these different parameters for the SPIO and the PHySIOMIC (table 1) and we added a paragraph in the discussion section, p21:

“These micro-sized particles enhance T_2^* contrast and enable rapid absorption for fast molecular imaging. Additionally, the polydopamine matrix conferred thrombosis targeting properties that enabled molecular MRI of microthrombi in a thrombo-embolic ischemic stroke murine model. It is also interesting to note that the PHySIOMIC provided strong imaging signal detection although they exhibited an extremely short half-life (less than a minute) compared to the SPIO which typically circulates much longer (over 10 minutes) but was not able to reveal the microthrombi. This might appear surprising as the field of nanomedicine usually supports long circulation half-life as a positive feature for molecular imaging applications as longer exposure time should logically result in increased targeting³⁵. However, this study provides evidence that this reasoning is not always true and that, at least in some contexts, modifying the surface to improve the targeting properties can be more valuable than to increase the circulation time.”

2. Figure 2b demonstrated that the imaging effect of PHySIOMIC was significantly superior to that of SPIO. It was proven that the protein corona of PHySIOMIC could enhance its microthrombi-targeting. However, there is no additional analysis of SPIO to illustrate the unique composition of the protein corona of PHySIOMIC. Please provide further evidence.

We have added the proteomic analysis of SPIO in the supplementary Figure S9. It shows that the amount of fibrinogen with the protein corona is lower that for the particles that bind to microthrombi (i.e. PHySIOMIC and melamine).

3. The formation of the protein corona on nanoparticles could accelerate the removal of particles from the bloodstream, thereby reducing the amount of PHySIOMIC reaching the microvasculature. It was also evident that the half-life of PHySIOMIC was much shorter than that of SPIO. Why did PHySIOMIC have a superior targeting to microthrombi? Please discuss the results.

Efficient targeting is indeed more important than long half-life to perform efficient molecular imaging and this study perfectly illustrates this fact. We believe that the composition of the protein corona and the size of the particles are two key aspects to target microthrombi. Moreover, the shorter half-life allows to perform imaging only a couple of minutes after intravenous injection, since plasma clearance

is very fast. We now raise this aspect in the discussion p 19:

It is also interesting to note that the PHySIOMIC provided a strong imaging signal detection although they exhibit an extremely short half-life (less than a minute) compared to the SPIO that typically circulate much longer (over 10 minutes) but were not able to reveal the microthrombi. This might appear surprising as the field of nanomedicine usually supports long circulation half-life as a positive feature for molecular imaging applications as longer exposure time should logically result in increased targeting³⁵. However, this study provides evidence that this reasoning is not always true and that, at least in some contexts, modifying the surface to improve the targeting properties can be more valuable than to increase the circulation time.

4. Figure 7a illustrated that both PHySIOMIC and SPIO particles induced a significant decrease in signal intensity in the liver and spleen, indicating rapid clearance of the particles from the body. Additionally, in Fig. 7b, it is noteworthy that the signal intensity after injection is considerably lower than before, and it gradually increases with time. Can you explain the underlying cause of this phenomenon?

We indeed attribute the strong signal decrease to rapid sequestration of the particles in the macrophages of the liver and the spleen. The iron oxide particles accumulate and confer negative signal. Then, the signal gradually increases with time once the iron oxide has been degraded and the iron incorporated into the iron pool of the organism. In the liver, it goes back to baseline at day 7 post injection.

We completed the result section p16:

“Signal enhancement was monitored in the liver, spleen and kidney at 2, 7 and 31 days after injection. Both PHySIOMIC and SPIO particles induced a strong hyposignal within the liver and the spleen (Fig. 7a) which is in line with a rapid elimination of particles from the bloodstream and uptake by the mononuclear phagocyte system; in the Kupffer cells of the liver and resident macrophages of the spleen.²¹ This hyposignal gradually fade out in time ending at baseline level at 31 days post injection. This observation was attributed to gradual biodegradation of the iron oxide particles within the lysosomal compartment of the macrophages; a mechanism well described in literature for inorganic nanoparticles, including iron oxide³³.”

5. The administration of thrombolysis with rtPA results in increased permeability of the blood-brain barrier. However, using PHySIOMIC at this stage is associated with an elevated risk of the contrast agent crossing the blood-brain barrier, which can potentially cause severe brain injury. Please discuss the advantages and disadvantages of PHySIOMIC as an MRI contrast agent.

We agree that this is another potential risk to consider in future developments. We added a paragraph in the discussion on that specific topic:

“Another risk to consider is the potential toxicity induced by PHySIOMIC particles that would eventually cross into the brain tissue, especially in the context of severe hemorrhagic transformation post-stroke. Although none of the current data did reveal PHySIOMIC retention other than the one bound to thrombi (due to their submicrometric size that prevents their passage through the blood-brain barrier (BBB)), the fate of the PHySIOMIC would need to be closely monitored for future development. Notably, a clear asset of this contrast agent is that it is fully made from biocompatible materials (iron oxide and polydopamine) that lack neurotoxic side effects. Accordingly, SPIO that can cross the injured BBB has been injected in humans at the subacute phase of ischemic stroke, without worrying signals for toxicity³⁹. Moreover, polydopamine nanoparticles exhibit strong antioxidant properties and were shown to result in a neuroprotective effect in an ischemic stroke model for this specific reason⁴⁰. These data are reinsuring for the clinical translation of PHySIOMIC.”

6. It is advisable to carefully proofread the manuscript for any errors in grammar and typos.

We thoroughly proofread the manuscript, corrected errors and typos and hope we did not miss any.

REVIEWERS' COMMENTS

Reviewer #1 (Remarks to the Author):

All concerns/suggestions addressed in a complete and detailed way.

No further concerns.

Reviewer #2 (Remarks to the Author):

It appears that the authors thoroughly addressed the vast majority of the reviewers comments resulting in a much stronger and highly impactful manuscript.

Reviewer #3 (Remarks to the Author):

My remarks were adequately addressed, the manuscript can be accepted.

Reviewer #4 (Remarks to the Author):

The authors have addressed the reviewer comments well and the manuscript is recommended for publication after careful checking of spelling and clarity of language in the text and captions.